# Rapid earthquake magnitude classification via P-wave strains from borehole strainmeters and Distributed Acoustic Sensing

T. M. Sawi [1] ✉, J. J. McGuire [1], A. J. Barbour [2], C. E. Yoon [3], M. Karrenbach[4] & C. Stewart[5]

Distributed Acoustic Sensing (DAS) offers a promising approach for earthquake early warning (EEW) in settings where seismic networks are costly to maintain. By repurposing fiber optic cables as dense strainmeter arrays, DAS enables real-time earthquake detection wherever those fibers are accessible. However, poor azimuthal coverage and challenges in estimating magnitude from strain measurements remain key hurdles in applying DAS to earthquake monitoring. Here, we develop a machine learning method to distinguish moderate-to-large (defined here as M ≥ 5.4) earthquakes from smaller ones within the first 4 sec of a strain waveform after a P-wave arrival without determining the earthquake location. Using ensemble decision tree models trained on borehole strainmeter data (3.5 ≤ M ≤ 7.1) and tested on onshore DAS waveforms (including the 2024 M7 Offshore Cape Mendocino earthquake), we find that low-frequency (0.2–0.5 Hz) continuous wavelet transform coefficients are the strongest predictors of magnitude, in addition to strain amplitude. Both DAS and borehole strainmeters effectively capture long-period strain signals, making these results valuable for EEW systems. Our method shows high precision compared to the real-time EEW system, ShakeAlert®, supporting the position that DAS is a viable technology for earthquake monitoring and magnitude classification. Fiber optic cables plus machine learning could aid earthquake early warning by estimating earthquake size within the first few seconds of shaking.

ShakeAlert® (hereafter referred to as ShakeAlert) is the earthquake early warning (EEW) system that provides seconds of warning of incoming seismic waves from moderate-to-large earthquakes to millions of people on the West Coast of the United States and Canada[1]. ShakeAlert uses the complementary detection capabilities of both seismic and geodetic sensors to accomplish this task. Many EEW algorithms rely on seismic stations to detect seismic activity in real-time, estimate the earthquake magnitude and location, and send alerts out to emergency systems, warning apps, and infrastructure managers[2–5]. The alerts grant time for citizens to take cover and allow for various automated preventative actions to take place, such as slowing trains and closing utility pipelines, thus directly contributing to the safety of the population[6].

[1]U.S. Geological Survey, Earthquake Science Center, Moffett Field, CA, USA. [2]U.S. Geological Survey, Earthquake Science Center, Vancouver, WA, USA. [3]U.S. Geological Survey, Earthquake Science Center, Pasadena, CA, USA. [4]Seismics Unusual, LLC, Fullerton, California, USA. [5]Office of the President, California State Polytechnic University, Humboldt, Arcata, CA, USA. ✉e-mail: tsawi@usgs.gov

However, there are high-hazard regions where seismic sensors are not in place, making EEW implementation more difficult and error prone[7]. For instance, at subduction zones, such as the Cascadia Subduction Zone (CSZ), offshore monitoring systems that are close to the subduction megathrust would help increase warning times, yet widespread submarine seismic networks are not currently operational due to the high costs of deployment, maintenance, and real-time telemetry[8–10]. This is a concerning weakness in the EEW system. Utilizing fiber optic sensing techniques such as distributed acoustic sensing (DAS) could potentially provide a means to fill this observational gap, as telecommunication companies operate underwater fiber optic cables in many locations, and DAS could effectively use them as strainmeter arrays with seismic-wave sensitivities[11–13].

DAS uses an interrogator at one end of the fiber that simultaneously transmits high-frequency pulses of laser light and measures the relative phase of Rayleigh backscattered light, typically from imperfections in the fiber. Deformation of the cable modifies the phase of the backscattered light, allowing interferometric measurements that can be converted to strain rate or strain via signal processing[14]. The last decade has witnessed a surge in studies of DAS for a variety of seismologic purposes[15–17]. By gaining access to offshore DAS cables, seismologists have the potential to monitor seismicity in subduction zones. This monitoring can be done in real-time via an onshore interrogator, thus potentially improving EEW in some of the most hazardous regions[9,18].

Understanding the utility of DAS in an EEW context is an active area of research[19–23], with recent studies inserting DAS measurements into existing EEW systems to expand the offshore range of the operational system[21,22]. A key criticism of submarine DAS cables, however, is that they tend to be geometrically linear arrays, which limits their azimuthal coverage and the accuracy of resulting location estimations. Additionally, the exact location of subsea cables can be uncertain, further reducing the reliability of location-based algorithms. This limitation is particularly worrisome for EEW, because offshore location errors tend to produce significant magnitude overestimates[9,18]. A key necessity for an offshore DAS-based EEW system is to improve warning times at coastal locations near the epicenter using only the earliest arriving *P*-wave data[7], as the algorithms using onshore seismic stations already perform well at warning more distant locations[18]. Additionally, many initial magnitude estimates utilize scaling relations that do not include finite fault effects[4,6] which could potentially bias magnitude estimates in M > 6 earthquakes. In this study, we bypass the need for estimating locations and instead directly characterize whether earthquakes near where the cable arrives onshore warrant an alert based on a binary magnitude classification scheme derived from the first four seconds of strain data after the *P*-wave arrival. We develop a machine-learning-based predictive model to identify key statistical descriptions that characterize moderate-to-large versus small earthquakes based on strain waveform data independent of knowledge of the earthquake location.

Our DAS data come from a 15-km-long DAS cable located in Humboldt County, California, onshore from the Mendocino Triple Junction at the southern end of the CSZ (ref. [24], Fig. 1a, b). Our predictive model was developed before the M7 December 5th, 2024, Offshore Cape Mendocino earthquake, and we hence hold out that event for testing the model. Our DAS dataset, therefore, contains only one M ≥ 5 earthquake (Fig. 1c), so we train our machine-learning models on statistical features derived from waveforms from seven regional borehole strainmeter stations from 2008–2018 from Mendocino County, Napa, and Ridgecrest, California, including 35 M ≥ 5 earthquakes (Fig. 2).

Our statistics include amplitude-based features as well as those derived from continuous wavelet transforms with various wavelet families filtered using a boxcar function over three different frequency bands (0.2–0.5 Hz, 0.5–2.0 Hz and 2.0–5.0 Hz; Fig. 3a). We theorize

that the lowest-frequency waveform features (0.2–0.5 Hz) may best characterize which earthquakes warrant alerts, as the longer ruptures in moderate-to-large earthquakes are reflected in the lower frequency content of the wavefield. Our four-second windows are long enough to capture the full duration of most M < 5.4 earthquakes and hence potentially differentiate them from larger, potentially damaging events.

Using these statistical features, we train a suite of 50 binary classification predictive models using a decision-tree-ensemble-based approach called XGBoost (ref. [25]; Fig. 4a, b) to predict whether the waveforms are generated by M ≥ 5.4 earthquakes. We choose a M5.4 cutoff because, at that threshold, ShakeAlert wireless emergency alerts reach a radius of 60 km around the estimated earthquake epicenter, as opposed to a M5, which only alerts out to a 25 km radius[18]. Thus, an offshore detection would likely include onshore alerts for an M ≥ 5.4. The objective is to give people a timely warning of dangerous shaking at distances where this is possible, therefore, the M5.4 threshold could be a logical choice for offshore DAS-based EEW to alert a predefined region near the cable landing. We ran the XGBoost model for 50 epochs to identify which features are most important for our binary prediction: "is the waveform from a M ≥ 5.4 earthquake?" We then find the number of XGBoost epochs (testing between 1 and 1000 epochs) to optimize the predictive scores. We apply the 50-epoch model, which was trained on strainmeter waveforms, to strain measurements of crustal earthquakes from a DAS deployment in Humboldt County, northern California (ref. [23]; Fig. 1a, b). By utilizing SHapley Additive exPlanations (SHAP) scores (ref. [24]; Fig. 4c), we rank and quantify how well each statistical feature used in the machine learning model contributes to correct predictions. We then evaluate the predictive model performance results using the average precision and recall of the suite of models (Fig. 4d) and compare our outcome to real-time ShakeAlert results. Finally, we calculate the most important predictive features for the December 5th, 2024, M7 Offshore Cape Mendocino earthquake on both DAS and borehole strainmeter data and use the existing model to predict whether waveforms from that event are greater than or equal to a magnitude 5.4, thus testing whether our model is a viable predictor on unseen data, including on unseen DAS data; an instrument on which the model has not been trained.

In this work, we characterize earthquakes of magnitudes 3.5 to 7.1 as either "large" (M ≥ 5.4) or "small" magnitude using a machine learning classifier trained on statistics derived from the first 4 s of strain waveforms after a *P*-wave arrival while utilizing no location information. We show how the machine learning model is transferable to DAS strain data and demonstrate that the "large" or "small" class of the earthquake is best predicted by low-frequency (0.2–0.5 Hz) continuous wavelet transform coefficients, in addition to strain amplitude. Both DAS and borehole strainmeters are ideal for capturing signals at this frequency band, and our method shows high precision compared to ShakeAlert®, providing empirical support that DAS is a viable technology for earthquake magnitude estimation and that the method described herein is a potentially valuable contribution to EEW systems.

## Results
### Machine learning outcomes: key features, model performance evaluation
Based on our average SHAP score rankings of the 50-epoch model runs of XGBoost, we find that 6 features ("features" being the scalar statistical values used as training input for the machine learning models) of the borehole strainmeter waveform data appear to stand out as most important for magnitude classification (that is, after the top 6 features, the SHAP feature importance score decreases, Fig. 5a). For further analysis, we use the 3 features which are less correlated with others to reduce redundancy among predictors and

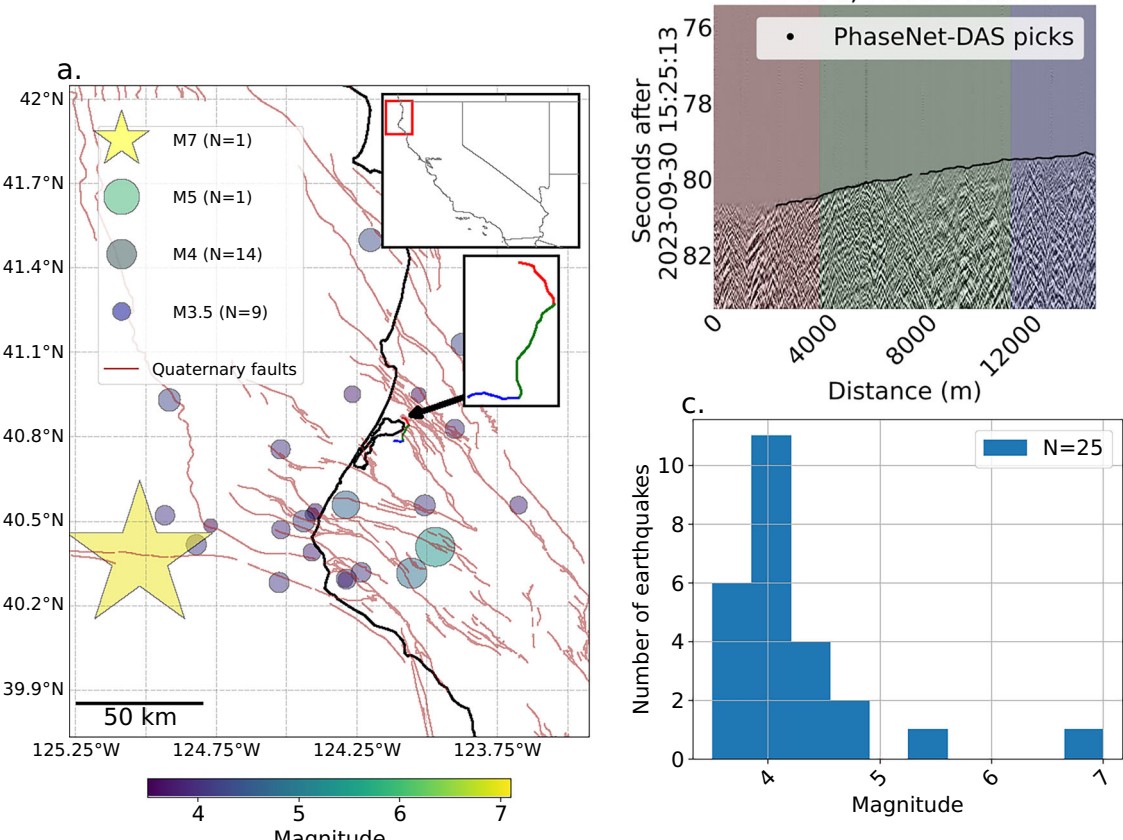

**Fig. 1 | Distributed acoustic sensing (DAS) earthquake dataset. a** Circles are M3.5+ earthquakes from December 2021- April 2023 within 100 km of the Eureka-Arcata cable in northern California. The star shows the hypocenter of the December 5th, 2024, M7 earthquake. The black arrow is pointing to the DAS fiber optic cable, featured in the lower inset figure. **b** Example waterfall plot for an M4.65 earthquake with *P*-wave picks (solid black line) determined by PhaseNet-DAS[24], with colors corresponding to cable segment colors shown in the inset figure in panel (**a**). **c** Magnitude distribution of M3.5+ earthquakes recorded on DAS.

avoid over-weighting strongly correlated information in the model interpretation. Those 3 remaining features are the maximum of the wavelet coefficients using 3rd order Gaussian wavelets filtered between 0.2-0.5 Hz, the range of the wavelet coefficients using 1st order Gaussian wavelets filtered between 0.5–2.0 Hz, and the maximum of waveform strain amplitudes (Fig. 5a, red text). The numbers following the Morlet wavelets in Fig. 5a refer to the width of the Gaussian envelope over the time domain (a higher number denotes a narrower envelope) and the frequency around which the wavelet's energy and the envelope are centered, respectively. All features are given in Supplementary Data 1, and their average ranks are given in Table S1.

After determining the most important features, we then systematically test between 1 and 1000 epochs to train XGBoost, finding that 500 epochs (with one model per epoch) lead to the optimal results (Fig. S5). Here, all training and testing are performed on borehole strainmeter data. We calculate the following mean performance metrics over the suite of 500 models: recall (i.e., of all waveforms from M ≥ 5.4 earthquakes, what percentage do we correctly predict?), precision (i.e., of all predictions of M ≥ 5.4 earthquakes, how many predictions are correct?), and F1-score (the harmonic mean of recall and precision). We calculate a mean accuracy of 0.96, but this score is inflated given the class imbalance of our dataset, i.e., the high number of M < 5.4 waveforms (N = 1847, or ~95% of the dataset) compared to M ≥ 5.4 waveforms (N = 102, or ~5% of the dataset; Fig. 5b). The mean recall, precision, and F1 scores of the 500 models are 81% (Fig. 5c), 79% (Fig. 5d), and 78%, respectively.

## Borehole strainmeter model performance comparison to ShakeAlert

We compare our machine-learning results with those of ShakeAlert, bearing in mind that our calculations are performed offline using 4 s after a manually picked *P*-wave arrival on strainmeter data, whereas ShakeAlert utilizes an automated, real-time *P*-wave detector to initiate most alerts and determines initial magnitude estimates with variable amounts of *P*-wave data ranging from about 0.5 to 5 s of data[4]. Our method also uses strain data from only one station without any location estimation, whereas ShakeAlert requires a minimum of detections at 4 seismic stations and must determine a location that is utilized in the magnitude estimation. Thus, any comparison involves significantly different datasets and associated limitations, and our results are not intended as a direct analog for the real-time system at this stage. Despite these differences, we present a comparison to ShakeAlert maximum real-time magnitude estimation results for 55 earthquakes, including 3 M ≥ 5.4 (ref. 26; Fig. 5e). This corresponds to all earthquakes between January 2021 and February 2024 within the ShakeAlert California reporting region with an estimated magnitude above M4.5. All three M ≥ 5.4 earthquakes are correctly predicted as such by ShakeAlert (Fig. 5f). However, ShakeAlert incorrectly predicted peak magnitude estimates for seven M < 5.4 earthquakes as M ≥ 5.4, resulting in a low precision score (30%; Fig. 5g). The retrospective results with our method have a higher precision: 79% of earthquakes predicted to be M ≥ 5.4 are in fact M ≥ 5.4 (Fig. 5d), compared to only 30% for ShakeAlert (Fig. 5g). Compared to ShakeAlert, our method may not correctly predict as many M ≥ 5.4 earthquakes (ShakeAlert alerts for 100% of M ≥ 5.4 earthquakes; Fig. 5f; compared to our 81%; Fig. 5c),

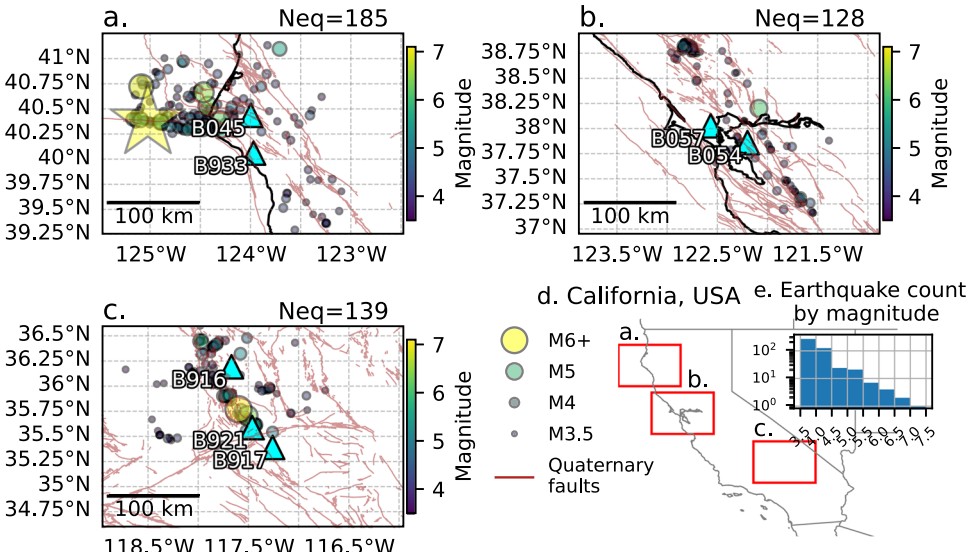

**Fig. 2 | Strainmeter earthquake dataset.** Circles are M3.5+ earthquakes from 2008–2022 within 100 km of borehole strainmeters in California (cyan triangles labeled with station names, thick black lines are coastlines), including those near **a** the Mendocino Triple Junction (star is December 5th, 2024, M7 earthquake), **b** Napa Valley, and **c** Ridgecrest. **d** California at the state-level with the study regions in (**a–c**) outlined in red. **e** Inset histogram displaying the number of earthquakes by magnitude in log-scale in 0.5-magnitude-unit bins.

but, in these simplified retrospective tests, our method is less likely to erroneously identify smaller earthquakes as larger ones that may warrant an alert.

## Comparing and predicting model performance on DAS data

For our analysis, we split the 15-km-long DAS cable into three segments based on cable geometry (red, green and blue boxes, Fig. 1b), calculating the statistical features for each channel then taking the mean and standard deviation for each of the three segments after removing low-sensitivity channels in the cable loops at points where the cable can be accessed for maintenance, leading to three data points for each of the 25 M ≥ 3.5 earthquakes recorded by the DAS data (see Methods). Figure 6a shows a strong correlation for both the borehole strainmeter and DAS data between the most important feature based on the SHAP scores (the maximum coefficient of the continuous wavelet transforms using the 3rd order Gaussian wavelets from 0.2–0.5 Hz), and maximum strain of the waveform in the first 4 s after the *P*-wave arrival, as well as a positive trend between higher values of those features and earthquake magnitude. Figure 6b shows a decline in strain amplitude with hypocentral distance. Importantly, the borehole strainmeter waveform's features do not cluster in 2-D space in Fig. 6 according to geographic region (i.e., Mendocino Triple Junction, Napa Valley, or Ridgecrest), demonstrating that these features are not site-specific. One notable exception to the distance-decay trend is the December 5, 2024, M7 Offshore Cape Mendocino event, which has a higher strain than expected given its 90–95 km hypocentral distance (red circles, Fig. 6b). This event ruptured west-to-east on a ~ 60 km-long fault[27], meaning that although the hypocenter was about 95 km away from the fiber, the rupture could have propagated approximately 15–20 km closer to the fiber during the first 4-s of rupture, (assuming an average rupture velocity of ~4.5 km/s[28]). This M7 event was not used in the training of the model, and when the 8 waveforms from that earthquake were tested (four channels each from 2 borehole strainmeter stations), all 500 models correctly predicted that all 8 waveforms were from an M ≥ 5.4 earthquake.

The starred points in Fig. 6 are the borehole strainmeter records where 140-s long duration waveforms exhibit static offset, as defined in Methods (see waveform examples in Fig. 7a). Their magnitudes range from M4.4 to M7.1, and their event-station distances range from 9.5 to

48 km (the maximum allowed distance being 50 km). Figure 6b shows how the smallest magnitude waveform examples with static offset also tend to be the closest to their respective stations.

We note a systematic offset between the data from the DAS and the borehole strainmeter instrumentations, regardless of distance: the borehole strainmeter data (Fig. 6a, x's) are overall higher in strain amplitude and have lower values in the continuous wavelet transform coefficient domain than the DAS data (Fig. 6a, squares). This offset persists even for the same earthquake (2024 M7 Offshore Cape Mendocino), which was recorded by borehole strainmeters (Fig. 6, red circles) and DAS (Fig. 6, red squares) in the same region. We attribute this offset to three factors related to the instruments and their installations: differences in burial conditions, site amplification effects between the hard rock housing the strainmeters and the soft sediment surrounding the DAS cable, and variations in the instruments' frequency responses. Figure 6b shows that for a given magnitude-distance, the DAS has a higher strain. Despite these data limitations, we test the statistical features derived from the 3 segments of the DAS cable from the Dec 5th, 2024, M7 event using the machine learning model and, across 500 testing iterations, achieve an average recall of 96%, demonstrating that for large earthquakes, a model trained to characterize earthquake magnitudes on borehole strainmeter data is suitable for DAS data as well.

We examine more carefully the most important feature for discriminating large from small earthquakes -- the maximum coefficient of the continuous wavelet transform using the 3rd order Gaussian wavelets from 0.2-0.5 Hz—specifically for the 2024 M7 Offshore Cape Mendocino event recorded on a representative channel of the DAS cable 94 km away, comparing it to a smaller (M4.8) event recorded on the same channel 33 km away. Figure 7a shows that both earthquakes register the same order-of-magnitude strain in the initial *P*-wave on the DAS channel and that the M7 waveform contains lower frequency content compared to the waveform from the M4.8 event. This variation in spectral content is evident not just in the waveform, but also in the CWT scalograms through time, which are shown as 4-s windows with 1-s overlaps (Fig. 7b, c), with a boxcar filter applied between 0.2–0.5 Hz. After the *P*-wave arrival, the M7 event (Fig. 7b) has higher CWT coefficient power than the M4.8 event (Fig. 7c), indicating that

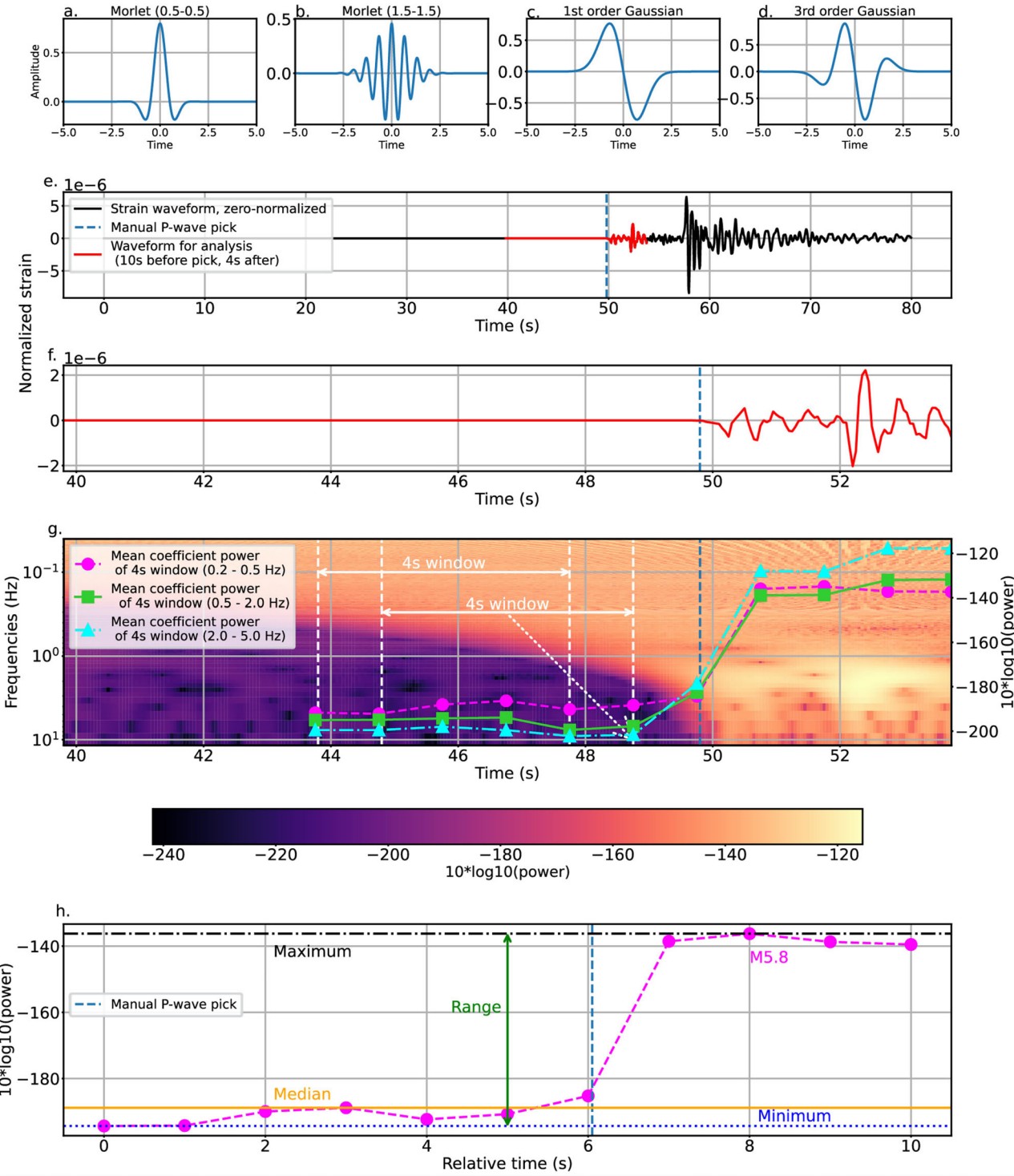

**Fig. 3 | Continuous wavelet transform (CWT) statistical feature calculation procedure. a–d** Four example wavelet families used in the CWT. The two numbers following the Morlet wavelets refer to the width of the Gaussian envelope over the time domain (a higher number denotes a narrower envelope) and the frequency around which the envelope is centered, respectively. **e** An 80-s strainmeter waveform containing a M5.8 earthquake with manual *P*-wave pick (blue vertical dashed line). **f** Zoomed-in section of 10 s before and 4 s after the *P*-wave pick used for analysis. **g** CWT scalogram using 3rd order Gaussian wavelet. We take 4-s sliding windows of the scalogram with 3-s overlap (white vertical dashed lines), then calculate the mean power (right vertical axis) across three different frequency bands (fuchsia circle dashed line (0.2–0.5 Hz), green square solid line (0.5–2 Hz), and cyan triangle dot-dashed line (2–5 Hz) markers). The mean values are calculated at the end of the 4-s windows, demarcated by the white dotted arrow. We connect the fuchsia (circle), green (square), and cyan (triangle) markers with dashed, solid, and dash-dotted lines, respectively, to indicate that they are now time series from which we will extract scalar values. **h** From these time series, we calculate the maximum, minimum, median, and range. One example time series is shown for the CWT scalogram (3rd Order Gaussian wavelet) mean power across the 0.2–0.5 Hz band (fuchsia circle dashed line); similar analyses were done for other frequency bands and wavelet families.

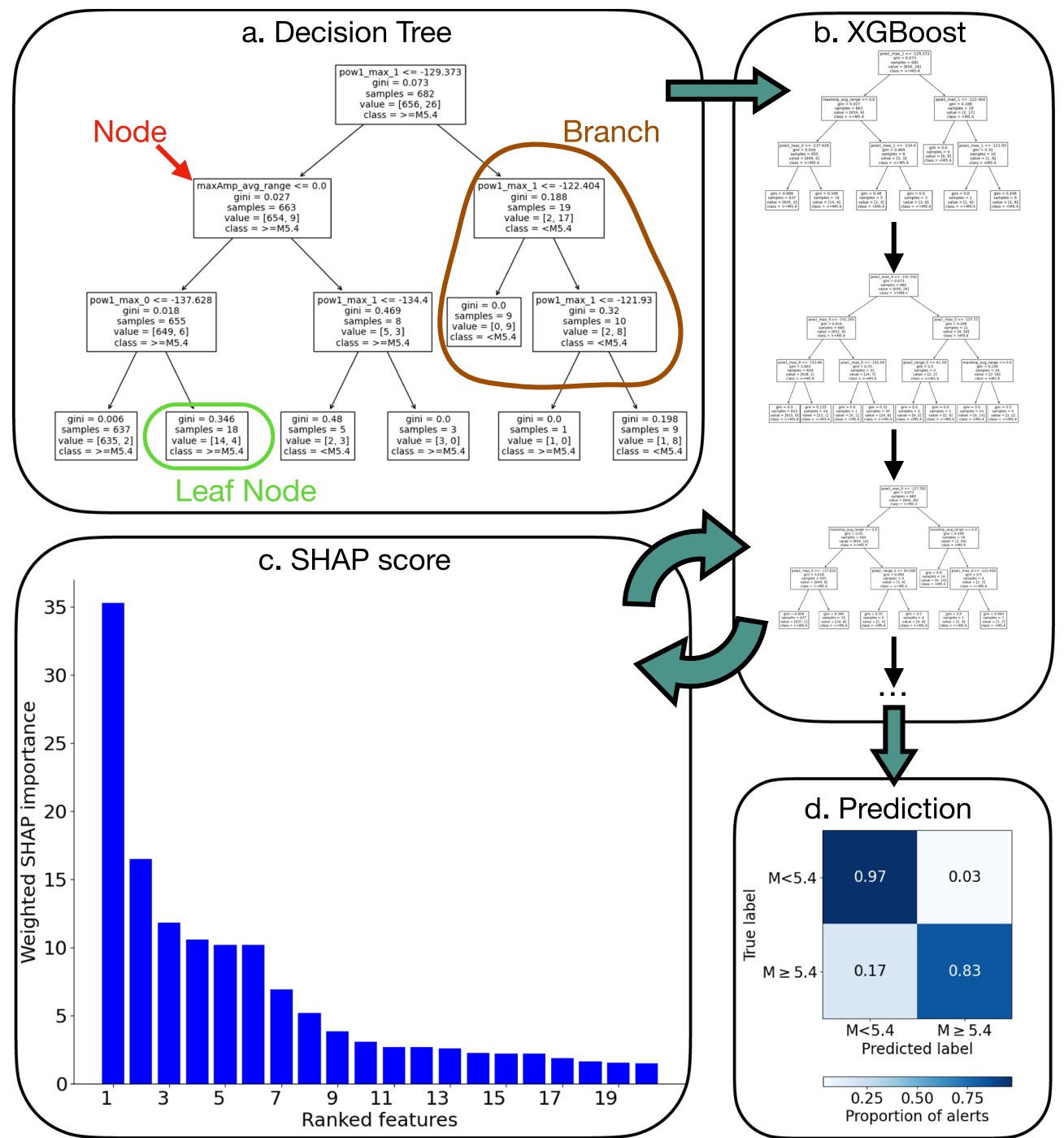

**Fig. 4 | Schematic of the machine learning workflow. a** A decision tree, in which each node selects a feature and a subset of training data, and partitions the data based on a binary decision. Additional implementation details are provided in the Supplementary Material. **b** XGBoost[25], which builds an ensemble of decision trees sequentially, with each tree learning from the errors of the previous ones. **c** Feature importance is evaluated using Shapley Additive exPlanations (SHAP values; ref. 26). Features are ranked using an inverse ranking scheme, and only the top-performing features are retained. **d** XGBoost is then re-trained on the reduced feature set to generate final predictions, which are evaluated using confusion matrices.

the lower frequencies are more prevalent in its waveform than in that of the smaller earthquake (likely resulting from a combination of a larger earthquake source duration and the increased attenuation on the longer propagation path of the M7 waveform).

**Static offset of borehole strainmeter waveforms**
Figure 8a shows 14 borehole strainmeter waveforms from M ≥ 4 events within 50 km of their respective stations exhibiting static offset in 140-s records. Our criteria for defining which waveforms contain a static

offset are given in Methods, and, in all cases, "maximum strain values" are defined as those found in the first 4 s after the P-wave arrival. Figure 8b demonstrates that the 4-s long records of those 14 waveforms (marked with stars) fall below the trendline (solid black line) fit to data (with each datapoint weighted by its maximum strain) up to maximum strain values of $10^{-6}$, then extrapolated up to a maximum strain of $10^{-4.5}$ (dashed black line). Indeed, large magnitude events in general tend to fall below the trendline in Fig. 8b. Likewise, in Fig. 8c, the trendline (black line), calculated up to maximum strain values of

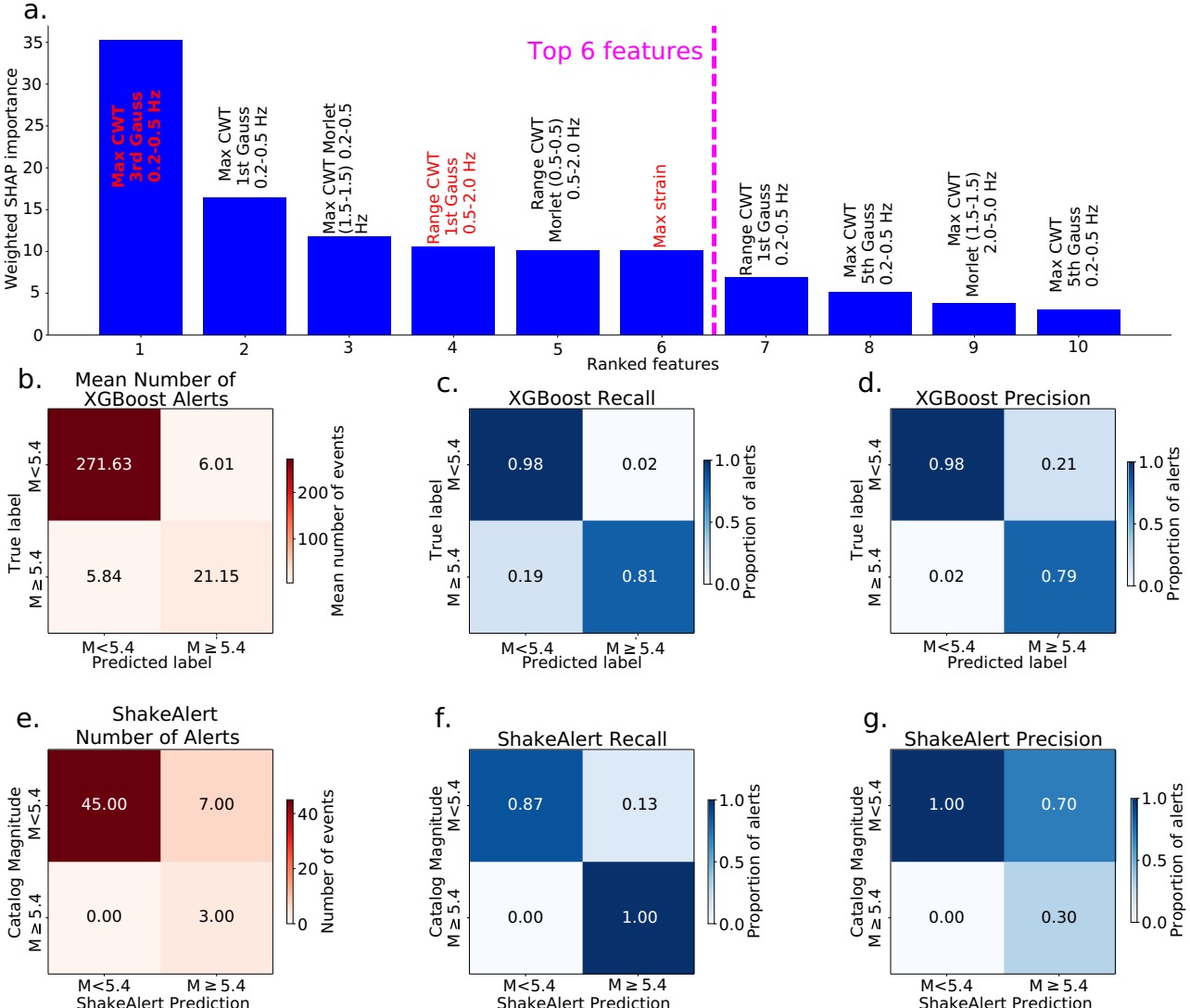

**Fig. 5 | Magnitude prediction results from XGBoost method compared to ShakeAlert. a** Ranked features from SHAP feature importance scores from the first round of XGBoost. The top 6 features (left of dashed fuchsia line) are determined from the second round of XGBoost. The 3 features which are less correlated with each other (those labeled with red text) are used in the final predictive model. **b**–**d** Confusion matrices from our XGBoost models averaged over 500 test sets. **b** Average number of events in a test set, **c** average recall, and **d** average precision.

**e**–**g** Same as (**b**–**d**) but comparing ShakeAlert's maximum magnitude predictions with ComCat catalog magnitudes used as ground truth[26]. **e** Number of events, **f** recall and **g** precision. ShakeAlert uses a variable amount of *P*-wave time window (-0.5–4 s) and a minimum of 4 seismic stations, but often many more. Unlike our method, ShakeAlert requires a location and magnitude estimate to predict expected ground motion intensity and operates in real-time. These results assume one alert per event.

$10^{-6.5}$ then extrapolated up to a maximum strain value of $10^{-4.5}$, fits DAS data well for low and moderate magnitude events, but the December 5th, 2024, M7 Offshore Cape Mendocino event falls well below the trendline.

## Discussion

Numerous studies have explored alternative approaches to applying machine learning for earthquake magnitude characterization. Early works utilizing convolutional neural networks on three-component seismograph data used waveforms that were tens of seconds long[29,30], thus limiting their utility in an EEW framework. Reference[31] used XGBoost on statistical features derived from 3 s of earthquake waveform data from single seismometers to predict magnitudes from global M3–M9 earthquakes, with the accuracy of results decreasing as magnitudes increased beyond M6. Similarly ref.[32], used XGBoost to predict the magnitude of moderate to large earthquakes on features extracted from strong motion seismographs, finding that low-

frequency features were some of the least important for predicting magnitude. Their waveform records, however, were filtered above 1 Hz, thus removing the long-period signals that we found essential for our magnitude characterizations. It would be difficult to establish that the frequency content of early arrivals has predictive power on the source magnitude without the ultra-broadband sensitivity of the strainmeters used in our training dataset.

More novel methods, such as Variational Mode Decomposition, which can denoise DAS and other seismic signals by compressing them into compact modal power spectral density features for deep learning, are gaining popularity[33,34]. However, their potential mode-mixing under non-stationary conditions and high computational latency may hinder real-time performance. Continuous wavelet transforms, on the other hand, provide adaptive frequency information and are ideally suited for non-stationary signals, such as earthquake waveforms. Meanwhile, machine-learning-enhanced real-time detection of prompt elastogravity signals (PEGS) could potentially grant warning times for

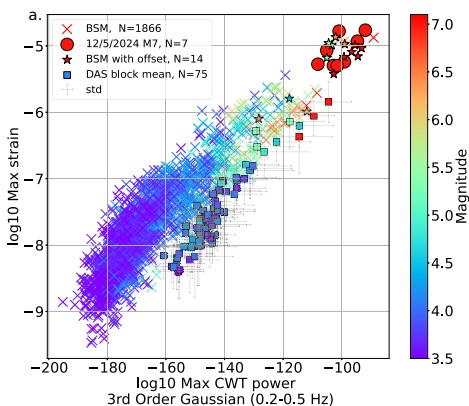
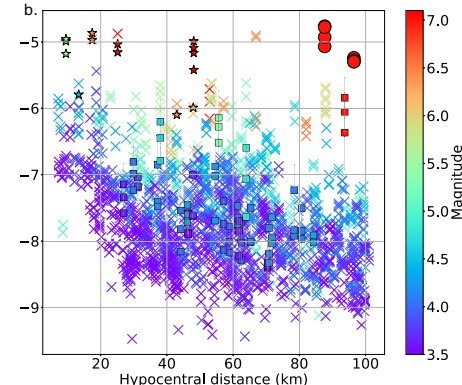

**Fig. 6 | Example statistical features for borehole strainmeter (BSM; x's) and distributed acoustic sensing (DAS) data (squares) colored by earthquake magnitude.** There are three squares per earthquake for the DAS data; one for each of the three segments of the cable (see Fig. 1b). Squares are the mean of the features calculated for all channels in the segment, and error bars show the standard deviation. Red circles and squares are borehole strainmeter and DAS data, respectively, from the December 5th, 2024, M7 Offshore Cape Mendocino earthquake. Stars are borehole strainmeter data showing static offset in the 140-s records (see Methods for our definition of static offset). **a** Maximum strain compared to the maximum CWT power for the 3rd order Gaussian wavelet family in the 0.2–0.5 Hz frequency band in the 4 s following the *P*-wave, the latter feature being found the "most important" feature for predicting whether the waveform came from an earthquake greater than or equal to magnitude 5.4. **b** Maximum strain compared to distance from the ComCat catalog earthquake hypocenter[26].

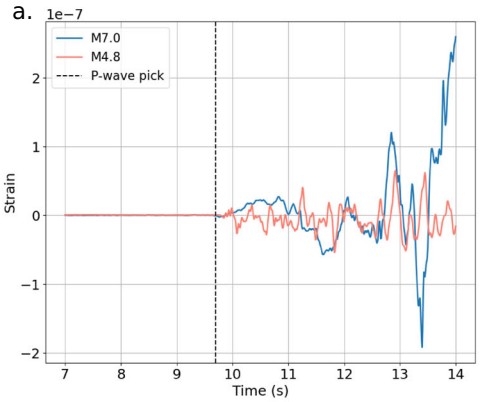
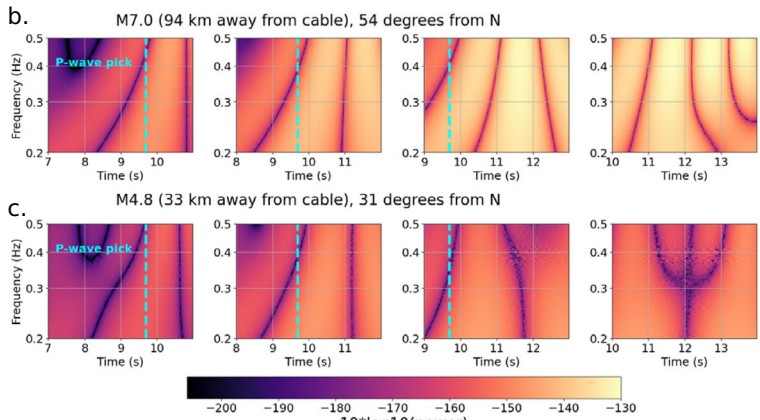

**Fig. 7 | Comparing M7.0 and M4.8 earthquakes recorded on a distributed acoustic sensing (DAS) cable, channel 2200 (of 3020).** "Zero seconds" is defined as 10 s before the *P*-wave arrival. **a** 7-s to 14-s unfiltered waveforms. **b**, **c** 4-s windows of 14-s records (with 1 s overlap) with continuous wavelet transform applied using 3rd order Gaussian wavelet. *P*-wave pick is at 4-s into the record, noted with a dashed cyan line. Color scales are the same for both the M7.0 (**b**) and M4.8 (**c**) earthquake records.

major earthquakes seconds to minutes faster than those from seismic instrumentation[35]. Observations of these signals, however, appear to be limited to M ≥ 7.9 earthquakes, making their application to EEW restricted to the largest earthquakes, and defining an important moderate-to-large magnitude gap that our method could fill. Additionally, the instrumentation required for the feasible use of PEGS for EEW systems is in the future development stages[36], whereas work incorporating low-latency, real-time DAS into operational EEW systems is already underway[22].

The top three most important features we found are all in the lowest tested frequency band: 0.2–0.5 Hz (Fig. 5a). We suggest that the successful performance of the low-frequency coefficients of the continuous wavelet transforms for identifying M ≥ 5.4 earthquakes from their earliest strain signals is due primarily to the scaling between magnitude, rupture extent, duration, and wavefield frequency content. Borehole strainmeters detect the long-period deformation from nearby large earthquakes, which may not be detectable above the noise level for nearby small to moderate-sized earthquakes[37].

Moreover, waveforms from large earthquakes recorded close to the fault are expected to show additional arrivals from near and intermediate field terms of the seismic wavefield that are potentially better recorded in the strain domain than with inertial sensors. More specifically, the near and intermediate field terms in strain correspond to propagating waves with amplitudes proportional to the cumulative seismic moment release or the moment-rate history and arrive coincident with the far-field *P*-wave term[38]. These terms could potentially enrich the low-frequency strain content of the initial *P*-wave, compared to the arrivals from smaller earthquakes. With proper coupling of the fiber optic cable, a DAS system should be able to measure these components of the wavefield[39].

The characteristic period ($\tau_p$), a low-frequency signal occurring in the seconds after the *P*-wave arrival, has been noted as a way to rapidly characterize the ultimate size of an earthquake[40,41], where a longer characteristic period would indicate a larger rupture length. Reference[42] estimated $\tau_p$ for the 2008 Mw 7.9 Wenchuan earthquake and aftershocks, finding that $\tau_p$ correlated well with magnitude but not

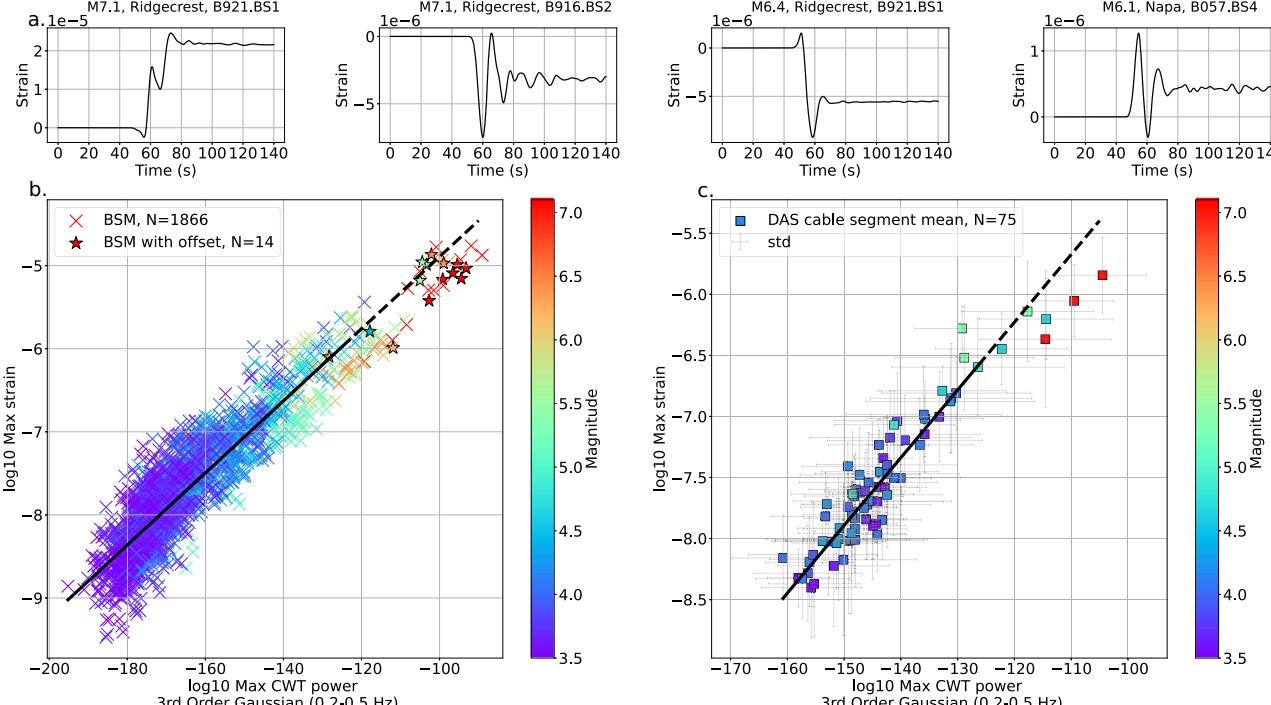

**Fig. 8 | Static offset and statistical features of borehole strainmeter (BSM) and distributed acoustic sensing (DAS) data. a** Four examples of BSM waveforms exhibiting static offset (all 14 waveforms with static offset shown in Fig. S6). **b**, **c** Maximum strain compared to the maximum CWT power for the 3rd order Gaussian wavelet family in the 0.2–0.5 Hz frequency band in the 4 s following the *P*-wave, the latter feature being found the "most important" feature for predicting whether the waveform came from an earthquake greater than or equal to magnitude 5.4 for (**b**) borehole strainmeter data, where stars indicate waveforms with static offset (see Methods for our definition of static offset), and **c** DAS data. DAS statistics are averaged across all channels along three 5-to-10-km-long segments of cable (see Fig. 1b, resulting in 3 points per earthquake), and error bars show the standard deviation within each cable segment. All data points are colored by earthquake magnitude. For panels (**b**, **c**), we fit a linear trend line (solid black line) between the minimum "Max strain" value and "Max strain" equal to $10^{-6}$ or $10^{-6.5}$ for panel (**b**, **c**), respectively, then extrapolate the trend line until the maximum "Max strain" value (dashed black line).

focal depth or epicentral distance, and when the *P*-wave was nodal, the correlation to magnitude became weaker. Similarly ref. 43, found that low-frequency wavelet coefficients from the first four seconds of the *P*-waves in the 2003 Tokachi-Oki M8.3 earthquake sequence recorded on near-field accelerograms showed scaling trends that could only be explained by the presence of the near- and intermediate-field components of the wavefield rather than just the far-field components. Moreover, the wavelet coefficients provided more differentiation between moderate and large magnitudes than simple peak-velocity or peak-acceleration scaling[43], mirroring our results from the strainmeter and DAS datasets (Fig. 7b, c).

Characterizing the spectral content may better distinguish whether a waveform comes from a large versus a small earthquake compared to the current practice in EEW (if near-field strain data can be well recorded via DAS). Current algorithms utilized by the ShakeAlert system combine measurements of peak displacement[6,44,45] or peak acceleration[2,3] combined with scaling relations for amplitude that fall off with distance from the epicenter or fault trace (Fig. 6b, ref. 46). These approaches work well when station geometries are uniform and surround the earthquake, but can lead to large errors for poor station geometries[7,18] or when the point-source approximation is not valid for a particular observation[46].

In Fig. 8b, c, some of the data points from larger (M ≥ 6) magnitude earthquakes are below the trendlines (black solid lines) fit using earthquakes with a maximum strain in the first four seconds after the *P*-wave up to $10^{-6}$ strain for the borehole strainmeter data, or $10^{-6.5}$ strain for the DAS data. Many of these data points (Fig. 7b, c, red) have larger CWT power coefficients for a given maximum strain value than would be expected by extrapolating the trendlines (black dashed lines). A key question is: what physical feature of the source or wavefield is

responsible for this offset from the trend that is exhibited by the higher strain events, but not the lower strain ones? We note that many of the waveforms showing static offset from events located within 50 km of the borehole strainmeters (waveforms in Fig. 8a and starred data, Fig. 8b, c) are below the trendline, as are the average statistical values of the DAS waveforms from the 2024 M7 Offshore Cape Mendocino event (Fig. 8c). We speculate that near or intermediate-field terms of the seismic wavefield may be responsible for the low-frequency content of the wavefield and contribute to the relatively high values of the CWT power coefficients compared to maximum strain of the waveform, but more detailed synthetic studies will be required to design wavelets, other statistical features, or filters intended to identify near and intermediate field terms in strain wavefields.

ShakeAlert currently uses magnitude estimates as key inputs to calculating ground motion estimates and alert decision criteria, but errors in both steps introduce large uncertainties for alert areas that are based on predicting ground motions[47]. Peak magnitude estimates are biased high in the current version of ShakeAlert in the M4-5.5 range[18], leading to the potential for over-alerting. Our method would not produce ground motion estimates to be used as alerting criteria, rather, it could supplement a standard EEW system like ShakeAlert with an additional "tripwire" like scenario for initial alerts in which large-magnitude, offshore earthquakes detected near a fiber optic cable would lead to the issuing of alerts over relatively contained and pre-defined onshore areas. This would be particularly useful for initial alerts along the coast, as waiting for onshore data often produces little to no warning time in those scenarios[18,48,49]. Now that we have identified strain features that perform well for discerning moderate-to-large from small magnitude earthquakes up to regional distances, our next goal is to test the machine-learning approach for large earthquake

magnitude classification on DAS data to determine its viability for issuing first alerts to regions near the cable landing site in real-time. Despite the large data volume associated with DAS, ref. 22 demonstrated the feasibility of integrating DAS data into an operational EEW system. Additionally, a real-time earthquake detection algorithm is already operational at the Arcata-Eureka cable used in this study[23]. Although time-optimization of our algorithms is beyond the scope of this work, one possibility is to compute feature extraction in parallel and on GPUs as part of the DAS data acquisition system with the machine-learning classification step onsite, thus working towards overcoming the constraints of real-time implementation[22].

One important uncertainty of our method is the unknown amplitude response of DAS fibers to nearby large magnitude earthquakes. Studies show that nearby large earthquakes can cause distortion of the DAS signal (that is, phase wrapping) depending on the proximity of the earthquake and the DAS instrument acquisition parameters[50]. In some cases, the true signal can be reconstructed with phase-unwrapping[51], and it may be possible to encode the presence of phase wrapping into the magnitude prediction model, assuming that an earthquake must be large, not small, if it is distorting the signal. Additionally, preventative measures can be taken by tuning interrogator acquisition settings to prevent phase wrapping: one can decrease the gauge length (in effect, making each channel less sensitive to strain changes), or increase the sampling rate, thus decreasing the phase change from one point to another[8]. That said, our DAS configuration has the dynamic range to accurately record moderate-magnitude local and regional earthquakes[23], including the initial P-waves of the 2024 M7 Offshore Cape Mendocino earthquake used in this study. Our method, however, focuses only on measuring the smaller-amplitude initial P-wave, not the S-wave, which would generally be less-prone to amplitude distortion and the fiber cable becoming decoupled from the solid Earth. Reference[8,52] noted the non-linear response of marine sediments in which subsea DAS cables are buried, which could amplify ground motions and possibly lead to liquefaction. Site-specific P-wave amplification is inherent to all DAS arrays with a significant span[53], and such amplification could complicate both the detection and magnitude characterization components of our method. Variable amplitude response can be measured along an offshore cable after sufficient earthquakes are recorded and subsequently encoded into the magnitude characterization scheme, potentially using the low amplification channels as recorders that will be less likely to distort in a large earthquake scenario. Practically, the cable can be split into segments according to P-wave amplification, then the statistical features can be scaled according to different amplification properties for each segment. Finally, our analysis has been carried out with onshore DAS cables or onshore borehole strainmeters, demonstrating that features in the 0.2–0.5 Hz range tend to perform best in terms of predictive ability. However, for offshore submarine cables, this is a frequency band that can be strongly influenced by swell noise, storm noise, near-shore currents, or flow noise (deep currents with small temperature gradients)[54,55]. These oceanic effects will depend on seafloor depths and other site-specific conditions. Detailed studies with submarine cables will be required to determine the extent to which these effects can be suppressed or eliminated with spatial array processing on DAS cable segments.

In this study, we reveal how low-frequency statistical features derived from borehole strainmeter waveform data within the first 4 s of a P-wave arrival from a single station, with no source location information, can be used to differentiate larger magnitude (M ≥ 5.4) earthquakes from smaller ones. We train a suite of machine learning models based on an ensemble decision tree method and discover that the most predictive features for rapidly differentiating larger earthquakes from smaller ones are the coefficients of continuous wavelet transforms filtered in low-frequency bands. These features act in essence as templates to match, and therefore detect, long-period

signals associated with the rupture process of nearby moderate-to-large magnitude earthquakes and are well suited to utilize the inherent advantages of the strain-wavefield compared to more standard seismic data. Despite a systematic offset existing between DAS and borehole strainmeter data in key statistical features (Fig. 6a), DAS data from the December 5th, 2024, M7 Offshore Cape Mendocino earthquake can be input into the suite of models trained on borehole strainmeters with excellent predictive results (in 96% of the 500 models, the M7 earthquake was correctly predicted as M ≥ 5.4). While the predictive model is successful for the range of source magnitudes tested in a variety of regions, including Ridgecrest, Napa, and onshore from the Mendocino Triple Junction in California, additional observations for M ≥ 5.4 earthquakes will be needed to confirm its robustness for larger magnitude events. Reference[56,57] applied deep learning on seismograms from Japan's Kyoshin network (K-Net) for rapid earthquake magnitude estimation, with both studies achieving robust results (e.g., a recall of 90.68% for M ≥ 5.5 events[57], compared to ours of 81% for M ≥ 5.4 events). The K-Net catalog has over 6000 M ≥ 5.5 seismometer records, compared to our borehole strainmeter catalog's 102 M ≥ 5.4 records, demonstrating the importance of large, balanced datasets for ensuring model accuracy. Both ref. 56, 57, however, utilize deep learning networks that conceal which features are most important for magnitude classification. Our model, on the other hand, is transparent in conveying the relative importance of features, allowing for better physical interpretation and practical application. Our method is generalizable across multiple regions in California, and consistency with outcomes found in ref. 58 suggest that our results are at least generalizable to the region covered by ShakeAlert (that is, the West Coast of the United States). Our method avoids calculating earthquake locations, making it a potentially important tool for characterizing earthquakes where relatively linear fiber optic cables delivering poor azimuthal coverage are the only instrumentation available, for example, in offshore subduction zone settings such as Cascadia and Alaska. Our work demonstrates that large earthquake magnitude binary classification with real-time DAS arrays may improve the initial EEW alerts in regions that are inaccessible with traditional seismic instrumentation.

## Methods
### DAS data
The Arcata-Eureka DAS array consists of a 15-km-long cable that spans from Eureka to Arcata, California, near the Mendocino Triple Junction at the southern end of the CSZ (Fig. 1a). The cable belongs to the fiber-optic-based internet provider Vero Networks and is rented by Cal Poly Humboldt, accessed via a Luna-OptaSense QuantX interrogator unit[23]. There are three periods of data collection: first, from May 18, 2022, to July 19, 2022, with a sampling rate of 250 Hz and a spatial interval of 2 m (7590 channels). These data were released publicly by the USGS[59,60]. The second period begins on December 22, 2022, as a rapid response to the December 20, 2022, M6.4 Ferndale earthquake with the same acquisition parameters as before and ends on February 3, 2023. This period captures most of the Ferndale aftershock sequence[61,62], including a M5.4 earthquake on January 1, 2023. The third data collection period goes from February 3rd, 2023, to April 2024, with new acquisition parameters of 100 Hz sampling rate and 5 m spatial sampling (3020 channels). All data are now publicly available[59,60]. To standardize the two data acquisition settings, we resample the two earlier 250-Hz data sets to 100 Hz. At the site of the interrogator, we run a real-time detection algorithm[23], which is accurate for local and regional events but insensitive to teleseismic ones.

Our DAS data set includes 25 M ≥ 3.5 earthquakes within 100 km of the array, including the M7 earthquake that occurred on December 5th, 2024, about 90 km offshore from Fortuna, California, near the Mendocino Triple Junction. We focus on local and regional earthquakes to evaluate whether nearby strain observations from only the first 4 s

after the earthquake *P*-wave arrival can help us discern the final size of the earthquake. We pick *P*-waves on the DAS data with PhaseNet-DAS (Fig. 1c), a deep-learning model trained to detect *P*- and *S*-wave arrivals on DAS data from Ridgecrest and Long Valley, California, as well as offshore Japan[63]. We repick missed *P*-waves, when necessary, by first splitting the cable into three ~5-10-km long segments based on cable geometry (red, green, blue in Fig. 1b), then picking approximate bounds in time around the arrival based on visual analysis and then rerunning PhaseNet-DAS within those bounds to obtain high-accuracy picks. We keep only earthquakes where an arrival can be visually confirmed, where the PhaseNet-DAS picks conform to that visual arrival, and where at least 10% of channels within the segment have picks. After utilizing PhaseNet-DAS, we apply a causal 4-pole Butterworth high-pass filter at 0.01 Hz to remove instrument noise and remove the baseline offset to each trace individually, thus zero-meaning the waveforms. Other than that, no signal processing is applied to the DAS data.

## Borehole strainmeter data

We focus on 3 regions where at least one large (M > 6) earthquake has been recorded between 2008 and 2022 on borehole strainmeters: Mendocino Triple Junction, Napa Valley, and Ridgecrest (Fig. 2a–c, respectively), all in California, USA. We collect strainmeter waveform data from the December 5th, 2024, M7 Offshore Cape Mendocino earthquake, but do not include them in our models' training steps. Instead, we use these data to test our resultant models to see if they can correctly predict these waveforms as originating from a large earthquake. We download all earthquake waveform data from the EarthScope Consortium Data Management Center (network code "PB", location "T0"; ref. 64), and preprocess the data (see Supplementary Information ref. 65), taking 40 s before and 40 s after the cataloged earthquake origin time. We only analyze earthquakes within a 100 km radius of any given strainmeter to focus on processes related to near-field strain deformation that can occur from large earthquakes' ruptures. We use all 4 channels (all horizontal for strainmeters), with a sampling rate of 20 samples per second, from 7 borehole strainmeters located throughout California (Fig. 2). The *P*-waves are manually picked on the strain data after applying a zero-phase Butterworth bandpass filter from 0.1 to 10 Hz. Waveforms that are too noisy or have uncertain *P*-waves are discarded, resulting in 1949 waveforms from 453 earthquakes with magnitudes from 3.5 to 7.1. Next, for the analyses on the first four seconds after the *P*-wave arrival, we use high-pass-filtered records of the same filter design above 0.01 Hz to minimize the impact of known noise sources; any remaining DC offset is removed by subtracting the mean value of the first 10 s of the data from the entire waveform. The waveform is trimmed from 10 s before the *P*-wave pick to 4 s after it. These 14-s waveforms are the only strainmeter data from which we extract our features, that is, our scalar statistical values, for our predictive modeling.

## Statistical feature extraction

We define feature extraction as the process of reducing the dimensionality of a dataset of vectors by computing scalar values from the vectors for the purpose of training a machine learning algorithm on those scalar values. Doing so allows for faster computation and less noise in the training stage than if using the full dimensionality dataset. For example, one could extract a feature from a waveform (normally a 1-D vector in strain amplitude) by computing its maximum strain (now a scalar value). We also extract features in the spectral domain via the utilization of continuous wavelet transforms (CWT)[66]. CWT scalograms are similar to spectrograms, except one may choose from several wavelet families for decomposing the waveform rather than just sines and cosines. Additionally, CWT has adaptive resolution, meaning the resultant scalograms have higher time resolution in high frequencies and higher frequency resolution in low frequencies. These are

desirable qualities for analyzing non-stationary and rapidly changing signals such as the arrival and passage of seismic waves.

Having extracted a 14-s waveform window of strainmeter data, we transform each window into time-frequency scalograms through continuous wavelet transforms using 10 wavelet families (Table S1; four example wavelets are depicted in Fig. 3a–d). We split the 14-s CWT scalogram (Fig. 3e, f) into 4-s windows with 3-s overlap, resulting in 11 windows filtered at three different bandwidths (0.2–0.5 Hz, 0.5–2 Hz, and 2–5 Hz; Fig. 3g) using a boxcar filter. Within each 4-s CWT scalogram window, for each of the three filter bands, we extract the mean power at the end of the 4 s (Fig. 3g). Thus, for each 14-s CWT scalogram, we then have three time series of mean power values (one per filter band), each time series containing 11 points from which we extract 4 scalar features: the minimum, maximum, median, and range (Fig. 3h). In addition, within each 4-s time window from the 14-s time-domain strainmeter waveform, we calculate the maximum strain, the range between the maximum and minimum of strain on linear and logarithmic scales, and three Boolean features: the maximum strain is greater than or equal to $10^{-5}$, $10^{-6}$, or $10^{-7}$. We then have, for each earthquake, for each station, and for each channel, a total of 127 features for each waveform (Supplementary Data 1). We calculate one more feature from the earthquake catalog: the target feature, a Boolean, "the earthquake catalog magnitude is greater than or equal to 5.4," and calculate the correlation between that target feature and each of the 127 statistical features for all waveforms (see Supplementary Data 1 for more details). We keep only the features with absolute correlations with the target feature that are greater than or equal to 0.5, resulting in 56 features that we will initially use to train the machine learning model to predict whether the waveform was generated by an earthquake with a source magnitude ≥ 5.4 (Fig. S1). We emphasize that this classification is based only on 10 s before the *P*-wave pick and 4 s after it, using only dynamic strain waveform content.

## Machine learning procedure

We utilize the XGBoost machine-learning model[25,67] to perform on the borehole strainmeter statistical features our binary prediction: is the waveform from a M ≥ 5.4 earthquake? XGBoost takes an ensemble approach, as it utilizes gradient boosting to sequentially train multiple decision trees (Fig. 4a) and each tree is improved by learning from the errors of the tree before it (Fig. 4b). The algorithm removes poorly performing submodels, that is, portions of trees, and implements regularization (details in Table S2), reducing model overfitting and making it a favored choice for online machine learning competitions[25,68]. We use a binary log-loss evaluation metric optimized for precision (that is, minimizing the number of false alarms), utilizing the "scale positive weight" option, which gives more weight to our large-magnitude earthquake class, as we have an order of magnitude fewer M ≥ 5.4 earthquakes than M < 5.4 earthquakes.

We split the features into 70%–30% randomly split training and testing sets, ensuring that all the waveforms generated from a single earthquake are contained in either the testing or training set, rather than being split between the testing and training sets. This is to prevent data leakage, as we assume that waveforms from the same earthquake will be mutually dependent. For EEW purposes, it is important to train the models to recognize large earthquakes while still allowing for enough large earthquakes to be in the testing sets. We have 42 strainmeter waveforms from large (M ≥ 6) earthquakes in our dataset (note that we do not include the December 5th, 2024, M7 earthquake in our initial test-training routine), and so impose the criteria that the training sets must have at least 15 of these large earthquakes but no more than 30. We initialize enough XGBoost models (~2000) with randomly split training-test sets to achieve 50 separate models that meet these criteria.

For the initial training round of XGBoost with 56 features, we grid-search with K-fold cross-validation to find the optimal

hyperparameters for our models. Details about the grid-search parameters can be found in Table S2, S4. We then determine which features are the most important for successful predictions by utilizing SHapley Additive exPlanations (SHAP) scores (ref. 24, Fig. 5a). SHAP scores explain how much each feature contributes to the expected model prediction during the training step, compared to if we did not use that feature. The SHAP score aggregates those contributions across the suite of 50 models to determine the relative importance of the features to the overall prediction. While XGBoost's internal gain metric indicates how much a feature improves the model's splits during training, it primarily reflects the feature's role in model construction rather than its direct influence on predictions. In contrast, SHAP values attribute changes in predicted scores to specific features, providing a more consistent and interpretable measure of their overall predictive contribution (Fig. S2). For this reason, we use SHAP scores to evaluate feature importance rather than the XGBoost gain metric.

We track the top ten most important features for each model epoch, then rank these features using an inverse ranking scheme to determine which features were most important for all models (Table S1). After the top 6 features (Fig. S3), we see a decrease in feature importance (Fig. 5a). We note that of those 6 features, 3 are highly correlated with each other (Fig. S3), so we remove the 3 features that are correlated and have lower SHAP scores, leaving 3 features (Fig. S4). We do one final round of XGBoost with just those top 3 features, again imposing that the training set must have at least 15 but no more than 30 waveforms from $M \geq 6$ earthquakes (grid search parameters found in Table S3). We implement rounds with 1, 10, 50, 100, 200, 500 and 1000 random train-test iterations, reporting the resultant average evaluation metrics (F1-score, recall and precision) for each number of iterations (Fig. S5). We find that after 500 iterations, the scores do not improve, so we choose 500 as our final number of model iterations for which we perform the binary predictions to average as our results (Fig. 5b–d), using the optimal hyperparameters found in the 56-feature model runs (Tables S3, S4).

### Defining static offset in strainmeter waveforms

We describe a final static offset for borehole strainmeter waveforms as the mean strain from 40 to 140 s after the manual *P*-wave pick, to minimize bias from the *S*-wave. The new baseline is defined as a static offset if two criteria are met: first, the absolute value of the new baseline must be more than $10^{-7}$ strain, and second, the absolute value of the new baseline must be more than 1.5 times the standard deviation of the entire waveform. These criteria were chosen by inspection to result in waveforms with a clear visual static offset (Fig. S6). In this way, the final static offset will not only be large in an absolute sense, but also relative to the dynamic wavefield portions of the waveform. Using this method, we find 14 waveforms ($4.4 \leq M \leq 7.1$) exhibiting static offsets (Fig. S6), some examples of which we show in Fig. 8a after a four-pole lowpass Butterworth filter at 1 Hz is applied to the waveforms, and which we indicate with stars in Fig. 8b.

### Data availability

The borehole strainmeter waveform data from the Plate Boundary Observatory (PBO) Borehole Strainmeter (BSM) network are openly accessible from the NSF NGF data archive operated by EarthScope Consortium (NSF award 2435260[64]). Data are available through the USGS data releases[59,60]. Source data are provided with this paper.

### Code availability

The code needed to recreate these results are available at https://gitlab.com/tsawi1/rapideqmag

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

## Acknowledgments

T. Sawi was supported by the U.S. Geological Survey Mendenhall fellowship program. Any use of trade, firm, or product names is for descriptive purposes only and does not imply endorsement by the U.S. Government. We thank Victor Yartsev for his assistance in acquiring the DAS data. This work benefited greatly from scientific discussions with the Powell Center Working Group on Fiber Optic Seismology for Earthquake Hazards Research, Monitoring and Early Warning.

## Author contributions

T.S – Conceptualization, Formal Analysis, Software, Visualization, Writing – Original Draft, Writing – review and editing; J.M. – Methodology, Supervision, Data Acquisition, Writing – review and editing; A.B. – Methodology, Supervision, Data Acquisition, Writing – review and editing; C.Y. – Methodology, Supervision, Writing – review and editing; M.K. – Data Acquisition, Methodology, Writing – review and editing; C.S. – Project Administration, Resources, Data Acquisition, Writing – review and editing

## Competing interests

The authors declare no competing interests.
