## [Transparent Peer Review file · Nature Communications]

Rapid earthquake magnitude classification via P-wave strains from borehole strainmeters and Distributed Acoustic Sensing

Corresponding Author: Dr Theresa Sawi

Version 0:

Reviewer comments:

Reviewer #1

(Remarks to the Author)

I read with great interest the manuscript entitled "Rapid earthquake magnitude estimation via P-wave strains from borehole strainmeters and DAS". The authors work, which is of high quality, explores an innovative approach to early earthquake warning using distributed fiber-optic sensing (DAS) data. By focusing on the frequency analysis of the first seconds of P-waves, the paper proposes an interesting automated method based on the XGBoost machine learning algorithm to distinguish moderate earthquakes from events likely to become very large, which has considerable potential in the field of seismic risk management. The topic is relevant, the results are original, and the approach could truly change how seismic alerts are handled, particularly in vulnerable yet isolated coastal areas.

That said, in my opinion, although the manuscript presents interesting advances, several points need to be addressed before considering publication in Nature Communications.

The first and most critical point concerns the geographical scope and, above all, the generalizability of the approach. The work is based on data specific to California and the ShakeAlert system. It would be important to explore in more detail the transferability of this method to other regions, especially those where DAS networks are less developed or calibrated, which ties into the authors stated objective. A discussion on the adaptability of the approach to other geophysical and technological contexts would benefit the study by making it more general and relevant to a broader readership.

Another important point concerns the practical application of this warning method. Although the authors clearly show that their approach could identify large earthquakes within the first seconds, a more concrete demonstration of its real-world implementation would be useful. What about the cost of continuous DAS data acquisition? Would it not be cheaper and maybe more relevant to densify coverage with additional stations rather than relying on tens of kilometers of DAS-interrogated cable? Automatisation is also not complete, as the authors repick some arrival after a first picking done by PhaseNet-DAS. What is the solutions here for an operationnal implementaiton ?

The instrumental limitations of DAS data, such as phase wrapping or limited dynamic range, deserve further discussion. The influence of oceanic forcings, particularly in the frequency range of interest (0.2–0.5 Hz), which overlaps with microseism peaks, is a critical point that needs to be quantified to assess the real-world applicability of the approach. This is mentioned in the discussion but warrants deeper analysis. It would be important to evaluate how these constraints might affect the accuracy of the proposed metrics and, consequently, the reliability of the warning system.

In my view, this article is interesting and carries genuine innovation with very good ideas, but it does not yet have the necessary maturity for a journal like Nature Communications. The project should move beyond the proof-of-concept stage by testing the system over longer periods and in different contexts (at least along the West Coast) to ensure its adoption by seismic risk operators and to have a real impact on how we manage this risk.

(Remarks on code availability)

Reviewer #2

(Remarks to the Author)

Thank you for your submission. The paper presents a machine learning based method using borehole strainmeter and DAS data to classify earthquakes with magnitudes above $M \geq 5.4$ by using the first four seconds of the P-wave signal. The study shows significant contribution to the scientific community as it advances the field of earthquake early warning (EEW) as well as using state-of-the-art technology as fiber-optic cables to expand the breadth of receivers used for EEW to areas where conventional sensors cannot access. An interesting point is that it bypasses the source location estimation by only detecting magnitudes, ignoring one of the main limitations of DAS when it comes to earthquake detection which is the aperture limitation. The manuscript is well written and it properly demonstrates its conclusions. Please see below specific comments.

- Were any conversions necessary for transferring the trained model to strainmeter to testing it on DAS? Was any normalization or scaling need before using the DAS data? The text notes differences in the data but it wasn't clear what was applied to the data.
- You mentioned the DAS dataset contains limited number of EQ, however, can you comment on the feasibility of using DAS for training as well, or hybrid training, could improve cross-domain generalization?
- Abstract: "within the first 4 seconds of a strain waveform after a P-wave arrival", consider changing to "first four seconds of strain waveform data" for smoother flow
- Page 2, line 66: You mention the limitation of using submarine DAS cable for hypocenter location, but the geometry/location of the cable itself can be highly uncertain as well, which can further reduce location reliability. You may consider addressing this as well.
- Page 2, line 58-59: "Vibrations in the cable allow (...)". I think this sentence is quite a simplification of how the method works, I'd suggest rephrasing it and improving for clarity. For example: "Strain and contraction of the cable modifies the phase of the backscattered light, allowing interferometric measurements that can be converted to strain rate or strain via signal processing."
- Line 418: "We define feature extraction as reducing the dimensionality...". Consider rephrasing to "We define feature extraction as the process of reducing..." for clarity.
- The model seems to predict well the M7 event, however further testing would be best to generalize and validate this outcome, so I suggest adding a clarification that future testing is needed on broader DAS dataset containing $> M5$ events to improve robustness.
- The SHAP analysis is an interesting addition to the analysis, could the authors comment whether SHAP importance rankings were consistent across different training/test splits?

(Remarks on code availability)

Reviewer #3

(Remarks to the Author)

Dear authors and editor,

Thank you for the interesting article. This study evaluates an algorithm for rapid magnitude classification between $M < 5.4$ and $M > 5.4$ using Decision Tree and XGB applied to borehole strainmeters and DAS data. The manuscript explains the methodology and presents its validation results, and I believe it should eventually be accepted. However, I suggest improvements to enhance clarity and make the study easier for readers to follow, as outlined below. I hope these suggestions will be useful for improving the presentation.

Major comments:

1. The current title suggests a continuous rapid earthquake magnitude estimation. However, the method described in the manuscript performs a binary classification to determine whether M is above or below a fixed threshold ($M=5.4$). I recommend revising the title to more accurately reflect the scope and output of the study.
2. The choice of the $M5.4$ threshold is not physically justified. In an earthquake EEWs, the magnitude threshold should be determined by the alert radius at which $PGA(M, d)$ can reach potentially dangerous levels. A proper approach is to define a shaking threshold (e.g., $PGA \geq 5\%g$ or a target MMI), compute the corresponding distance for different magnitudes using an appropriate regional GMPE, and select the minimum magnitude whose shaking footprint covers the target area. The wireless emergency alert should then be configured to match this hazard-based footprint, rather than selecting the magnitude to match the operational alert radius of wireless emergency alerts (for example, <https://doi.org/10.1785/0120240119>, <https://doi.org/10.5194/nhess-20-921-2020>).
3. Given that the five most significant features are highly correlated (Fig. S2, with three of them being 100% correlated). Why not removing these attributes to improve the predictions?

Minor comments and questions:

1. In the abstract: "shows high precision compared to real-time EEW systems." Specify which EEWs you have compared with; in this case, it is ShakeAlert. The reading flow even starts with ShakeAlert in the introduction.
2. Line 93: Perhaps add ref. Meier et. al 2017 (<https://doi.org/10.1126/science.aan5643>), because the Source Time Function for large earthquakes is typically < 6 s.
3. Line 96-99: Which hyperparameters were used for the Decision Tree and XGB models? For example, number of trees, maximum depth, percentage of the initial feature vector, etc. You might also consider adding a reference that uses a single station, XGB, and 3 seconds of the P-wave (<https://doi.org/10.1029/2023JB026575>).
4. In the feature importance analysis, I wonder if the SHAP scores are highly correlated with the feature importance values

provided by the XGB model.

5. Line 133: It is unclear what exactly constitutes these 500 models. Are these 500 models trained with different parameters? Please clarify whether this refers to 500 independent training runs (e.g., via cross-validation, bootstrapping, or parameter variation).

6. Line 160: True Positive (TP) is defined as $M > 5.4$ earthquakes estimated as $M > 5.4$, FP as $M < 5.4$ earthquakes estimated as $M > 5.4$, and FN as $M > 5.4$ earthquakes estimated as $M < 5.4$. The strong point is the larger number of $M > 5.4$ samples used in the proposed algorithm compared to only 3 in ShakeAlert. The precision and recall for these data indicate that the proposed algorithm has a lower chance of producing an FP. However, in an EEW, the critical issue is the algorithm's failure in the case of an FN.

(Remarks on code availability)

Version 1:

Reviewer comments:

Reviewer #1

(Remarks to the Author)

Dear Authors, dear Editors,

I have read this manuscript with great interest. I think this is an excellent and scientifically very robust manuscript that unquestionably deserves publication. Its technical quality, clarity, and rigor are high, and the results represent a meaningful contribution to the field. However, as currently written, the manuscript does not articulate the kind of broad, field-shaping impact expected for Nature Communications. The work may be more appropriately aligned with a high-level disciplinary journal (such as Nature Geoscience, Communications Earth & Environment, or GRL), where its strengths would be fully appreciated.

The methodological approach relies on established techniques applied to a familiar classification problem, and the central findings, while interesting, do not emerge as a clear conceptual advance that would reshape current understanding or practice. The physical interpretation remains qualitative, and the potential implications for early source characterization are not explored in a way that convincingly highlights a transformative contribution.

The discussion of DAS-based applications would benefit from a more realistic appraisal of current operational limitations. DAS remains extremely challenging to process in real time at the scale required for EEW, and to date, to my knowledge, it is not used operationally for monitoring precisely because of the computational burden and data volume. Without demonstrating real-time feasibility or proposing a strategy to overcome these constraints, it is difficult to evaluate the practical significance of the DAS extension. A second concern relates to the question of model transferability. The manuscript focuses on a single region and dataset, yet does not explore whether the proposed features or learned representations would generalize to other tectonic contexts, sensor networks, or noise environments.

The positioning of the work would also benefit from a deeper engagement with the most recent literature. There is a notable gap in the manuscript's coverage of the broader Earthquake Early Warning (EEW) literature, particularly work that explicitly targets rapid magnitude estimation and waveform classification using machine-learning approaches. The authors do not sufficiently contrast their findings with key recent studies (e.g., the line of work following Lomax et al., 2019, and related developments) nor with methodological advances coming from adjacent fields, such as rapid frequency-domain classifiers or approaches inspired by gravity-waves signal processing. It would strengthen the paper significantly if the authors (i) positioned their contribution within these bodies of work, (ii) clarified what is genuinely new in terms of methodology or physical insight compared with these precedents, and (iii) provided direct comparisons when possible or justified their absence.

I think that this study, while promising, does not yet demonstrate the level of maturity expected for a Nature Communications contribution. Showing that the approach can scale beyond a single region, operate in real-time conditions, and consistently perform on a broader set of events would considerably strengthen the case for publication in a journal with the wide reach and expectations of Nature Communications. Once such demonstrations are in place, the work may well achieve the level of generality and impact required for that venue.

(Remarks on code availability)

Reviewer #2

(Remarks to the Author)

I believe my comments have been addressed.

(Remarks on code availability)

Reviewer #3

(Remarks to the Author)

Thank you for considering my suggestions. I have no further comments to make. I am looking forward to seeing the paper in its final format. Congratulations.

(Remarks on code availability)

Version 2:

Reviewer comments:

Reviewer #3

(Remarks to the Author)

Thank you for your interest in my opinion regarding the critical points raised. I feel that the authors have adequately addressed the comment concerning model transferability. However, I would like to offer one additional suggestion that could further strengthen the discussion of real-time feasibility.

The authors have now added references to DAS-based EEW systems in the Discussion section, which reinforces the case for future implementation of their methodology. It may be helpful to explicitly mention that the USGS is actively working toward implementing DAS in real time on a nationwide scale, so that readers clearly understand that DAS-based EEW is likely to become an operational reality.

To further support the feasibility of the proposed methodology within an EEW framework, the authors could consider including an estimate of the algorithm's time complexity, excluding data transmission and telemetry delays. For example, this would consist of the required 4 seconds of P-wave data plus the computational time needed for feature extraction and classification. Even more compelling would be to provide an estimate of the expected user lead time, calculated as the difference between the theoretical S-wave arrival time (or the time corresponding to observed 5%g shaking) at a given location and the time at which the algorithm classifies the event as large (see the Discussion section on the "late-alert zone" in <https://doi.org/10.1785/0120240119>).

In the first paragraph of the Discussion, the authors may also consider citing reference (49), Lara et al. (2023), including its real-time application (<https://doi.org/10.1785/0120240119>), as this methodology likewise estimates large earthquakes using XGBoost with waveforms filtered above 1 Hz.

I would recommend acceptance of the manuscript in its current form. Nevertheless, I believe that incorporating these clarifications would further strengthen the paper, particularly with respect to the real-time feasibility considerations raised during the review process.

(Remarks on code availability)

REVIEWER COMMENTS

We appreciate the thorough and thoughtful comments that all three reviewers left for our manuscript. We found them extremely helpful and have addressed them to the best of our ability. The reviewers' words are in their original form, nothing has been removed, but the questions/comments that required responses are numbered and bolded. Our responses are in blue.

Comments from Reviewer #1:

The first and most critical point concerns the geographical scope and, above all, the generalizability of the approach. The work is based on data specific to California and the ShakeAlert system. It would be important to

- 1. explore in more detail the transferability of this method to other regions, especially those where DAS networks are less developed or calibrated, which ties into the authors stated objective.**

We believe the following sentence addresses this concern; note we have removed the claim that it may be transferrable globally (Discussion, last paragraph) "...and consistency in outcomes found in (48) suggest that our results are at least generalizable to the region covered by ShakeAlert (the West Coast of the United States).

48. A. J. Barbour, J. O. Langbein, and N. S. Farghal, "Earthquake Magnitudes from Dynamic Strain," *Bull. Seismol. Soc. Am.* 111, 1325–1346 (2021). <https://doi.org/10.1785/0120200360>

- 2. A discussion on the adaptability of the approach to other geophysical and technological contexts would benefit the study by making it more general and relevant to a broader readership.**

We do think the methodology would be adaptable to most/all borehole strain and DAS/fiber datasets with sufficient dynamic range. Our borehole dataset covers a variety of tectonic environments. There's nothing particularly unusual about our DAS dataset except perhaps the dynamic range being somewhat higher than studies that use longer cables and lower ping rates. However, this is not a large problem in that we only use the first few seconds of P-wave data which have relatively small amplitudes. We have expanded the discussion of this point in section 3 (page 9) on this point. It is not referenceable yet, but current prototype DAS interrogators are able to provide high quality data using only 2-meter gauge lengths out to ~75 km distance. This will effectively eliminate much of the

saturation problem for P-waves as these instruments come online in the near future.

In terms of other measurement technologies perhaps the most comparable to strain sensors are traditional geodetic data. Real-time GNSS data is quite noisy and currently limited to magnitudes above about M7.0 [Murray et al., 2024]. Strain measurements are fundamentally different than inertial measurements (e.g. ground acceleration or velocity) and hence we're not exactly sure which technological contexts you're referring to.

Murray et al:

<https://doi.org/10.1785/0120220181>

Another important point concerns the practical application of this warning method. Although the authors clearly show that their approach could identify large earthquakes within the first seconds, a more concrete demonstration of its real-world implementation would be useful. What about the

- 3. cost of continuous DAS data acquisition? Would it not be cheaper and maybe more relevant to densify coverage with additional stations rather than relying on tens of kilometers of DAS-interrogated cable? Automatisation is also not complete, as the authors repick some arrival after a first picking done by PhaseNet-DAS. What is the solutions here for an operationnal implementaiton ?**

A detailed cost analysis is beyond the scope of this study and a moving target. The cost of DAS instrumentation is dropping rapidly and long term operation and maintenance costs may already be much lower than that of a seismometer array that stretches tens of kilometers while delivering a much denser sensor spacing. The newest interrogator models can take data on at least four cables simultaneously allowing centralized data collection for a wide region without a telemetry network. The cost of station maintenance and telemetry must be considered, and even then, many remote seismic stations will experience more downtime than the DAS cable. The California Integrated Sesimc Network has already replaced some stations that experience periodic vandalism with a few DAS channels. A study of DAS completeness levels is being published elsewhere by a USGS Powell Center working group. Completeness levels >99% for over a year without a site visit are now common. That study and a related ANSS working group are looking at ways to reduce long-term archiving costs which are a significant issue for DAS but likely surmountable with tiered retention schemes and compression algorithms. In short, the cost comparison is complex and changing rapidly and beyond the scope of this paper.

We have a real-time detection algorithm in operation onsite at the Arcata-Eureka DAS cable that uses temporal variations in spectral power. We plan to implement this method in tandem with a P-wave picker in the near future, but not before publication of this manuscript. We've added the line accordingly (section 4.1, paragraph 1):

4. **The instrumental limitations of DAS data, such as phase wrapping or limited dynamic range, deserve further discussion.**

Added to discussion: “Additionally, preventative measures can be taken by tuning acquisition settings to prevent strain saturation on a DAS array. One can decrease the gauge length (in effect, making each channel less sensitive to strain changes), or increase the sampling/ping rate, thus decreasing the phase change from one time point to another (Mellors et al., 2021). That said, DAS data has the dynamic range to accurately record moderate local and regional earthquakes (Yin et al., 2023; Strumia et al., 2024), including the 2025 M7 Offshore Cape Mendocino earthquake shown in this study, 100 km away from the DAS cable.”

5. **The influence of oceanic forcings, particularly in the frequency range of interest (0.2–0.5 Hz), which overlaps with microseism peaks, is a critical point that needs to be quantified to assess the real-world applicability of the approach. This is mentioned in the discussion but warrants deeper analysis. It would be important to evaluate how these constraints might affect the accuracy of the proposed metrics and, consequently, the reliability of the warning system.**

We have not analyzed any submarine cable datasets and can't address this topic in that setting directly in this paper. Our DAS cable is located very close to the coast, even in a salt marsh at one point, in a region of constant large swell and strong winter storms. Many of the DAS points in Figures 6-8 are from earthquakes during the winter of 2022-2023 and contain strong microseism signals already, albeit on a terrestrial cable. For the EEW application, we are mostly interested in large signals from earthquakes near

the cable and we expect the method could be adjusted for a particular submarine cable.

Jiaqi Fang, Yan Yang, Zhichao Shen, Ettore Biondi, Xin Wang, Ethan F. Williams, Matthew W. Becker, Dominic Eslamian, Zhongwen Zhan; Directional Sensitivity of DAS and Its Effect on Rayleigh-Wave Tomography: A Case Study in Oxnard, California. *Seismological Research Letters* 2022;; 94 (2A): 887–897.
doi: <https://doi.org/10.1785/0220220235>

In my view, this article is interesting and carries genuine innovation with very good ideas, but it does not yet have the necessary maturity for a journal like Nature Communications. The project should move beyond the proof-of-concept stage by

- 6. testing the system over longer periods and in different contexts (at least along the West Coast) to ensure its adoption by seismic risk operators and to have a real impact on how we manage this risk.**

We believe this sentence addresses this concern (Discussion, last paragraph) “and consistency in outcomes found in (48) suggest that our results are at least generalizable to the region covered by ShakeAlert (the West Coast of the United States).

48. A. J. Barbour, J. O. Langbein, and N. S. Farghal, "Earthquake Magnitudes from Dynamic Strain," *Bull. Seismol. Soc. Am.* 111, 1325–1346 (2021). <https://doi.org/10.1785/0120200360>

Reviewer #2 (Remarks to the Author):

Thank you for your submission. The paper presents a machine learning based method using borehole strainmeter and DAS data to classify earthquakes with magnitudes above $M \geq 5.4$ by using the first four seconds of the P-wave signal. The study shows significant contribution to the scientific community as it advances the field of earthquake early warning (EEW) as well as using state-of-the-art technology as fiber-optic cables to expand the breadth of receivers used for EEW to areas where conventional sensors cannot access. An interesting point is that it bypasses the source location estimation by only detecting magnitudes, ignoring one of the main limitations of DAS when it comes to earthquake detection which is the aperture limitation. The manuscript is well written and it properly demonstrates its conclusions. Please see below specific comments.

- Were any conversions necessary for transferring the trained model to strainmeter to testing it on DAS? Was any normalization or scaling need before using the DAS data? The text notes differences in the data but it wasn't clear what was applied to the data.

The model was not altered before being used to test the DAS data. No scaling or normalization was applied to the DAS data. The strain data from DAS was similar enough the strain data from the boreholes for both to be tested by the same model. We've now included as the last sentence of section 4.1:

"After utilizing PhaseNet-DAS, we apply a causal 4-pole Butterworth high-pass filter at 0.01 Hz to remove instrument noise and remove the baseline offset to each trace individually, thus zero-meaning the waveforms. Other than that, no signal processing is applied to the DAS data."

- You mentioned the DAS dataset contains limited number of EQ, however, can you comment on the feasibility of using DAS for training as well, or hybrid training, could improve cross-domain generalization?

There is only one nearby $M \geq 5.4$ earthquake in the DAS dataset (we were holding out the M7 for testing), meaning that the DAS dataset was class-imbalanced and inadequate for training. We expect a future study can attempt this as the dataset grows.

- Abstract: "within the first 4 seconds of a strain waveform after a P-wave arrival", consider changing to "first four seconds of strain waveform data" for smoother flow

It is important to emphasize that it's after the P-wave arrival

- Page 2, line 66: You mention the limitation of using submarine DAS cable for hypocenter location, but the geometry/location of the cable itself can be highly uncertain as well, which can further reduce location reliability. You may consider addressing this as well.

Added after the line mentioned above: "Additionally, the exact location of a subsea cable can be uncertain, further reducing the reliability of location based EEW algorithms."

- Page 2, line 58-59: "Vibrations in the cable allow (...)". I think this sentence is quite a simplification of how the method works, I'd suggest rephrasing it and improving for clarity. For example: "Strain and contraction of the cable modifies

the phase of the backscattered light, allowing interferometric measurements that can be converted to strain rate or strain via signal processing.”

Changed the wording as suggested

- Line 418: “We define feature extraction as reducing the dimensionality...”. Consider rephrasing to “We define feature extraction as the process of reducing...” for clarity.

Changed wording as suggested

- The model seems to predict well the M7 event, however further testing would be best to generalize and validate this outcome, so I suggest adding a clarification that future testing is needed on broader DAS dataset containing > M5 events to improve robustness.

Added line : “ While the predictive model appears to be robust for the range of source magnitudes tested here, additional events with $M \geq 5$ will be needed to confirm its robustness for larger magnitude events, especially in different crustal and tectonic regimes.”

- The SHAP analysis is an interesting addition to the analysis, could the authors comment whether SHAP importance rankings were consistent across different training/test splits?

The text mentions that the SHAP scores are averaged across the test and training splits, so one can assume that they are similar enough to give the average results.

Reviewer #3 (Remarks to the Author):

Dear authors and editor,

Thank you for the interesting article. This study evaluates an algorithm for rapid magnitude classification between $M < 5.4$ and $M > 5.4$ using Decision Tree and XGB applied to borehole strainmeters and DAS data. The manuscript explains the methodology and presents its validation results, and I believe it should eventually be accepted. However, I suggest improvements to enhance clarity and make the study easier for readers to follow, as outlined below. I hope these suggestions will be useful for improving the presentation.

Major comments:

1. **The current title suggests a continuous rapid earthquake magnitude estimation. However, the method described in the manuscript performs a binary classification to determine whether M is above or below a fixed threshold ($M=5.4$). I recommend revising the title to more accurately reflect the scope and output of the study.**

New title: Rapid earthquake magnitude classification via P-wave strains from borehole strainmeters and DAS

2. **The choice of the $M5.4$ threshold is not physically justified. In an earthquake EEWS, the magnitude threshold should be determined by the alert radius at which $PGA(M, d)$ can reach potentially dangerous levels. A proper approach is to define a shaking threshold (e.g., $PGA \geq 5\%g$ or a target MMI), compute the corresponding distance for different magnitudes using an appropriate regional GMPE, and select the minimum magnitude whose shaking footprint covers the target area. The wireless emergency alert should then be configured to match this hazard-based footprint, rather than selecting the magnitude to match the operational alert radius of wireless emergency alerts (for example, <https://doi.org/10.1785/0120240119> , <https://doi.org/10.5194/nhess-20-921-2020>).**

The primary goal of EEW systems is to deliver useful warning times, not precise ground motion predictions. For instance, while the U.S. ShakeAlert system's products are formally defined as ground motion predictions, USGS has recognized that this is an unrealistic goal near the epicenter and has codified an increased tolerance for overalerting within 100km of the epicenter estimate [McGuire et al., BSSA 2025]. Moreover, the USGS actively advises implementation organizations (such as cellphone OS companies, trains, ..) to not use any ground motion thresholds higher than MMI 3.5 because the alerts will be too slow in many cases [McGuire et al., 2025]. In the current ShakeAlert implementation, MMI 3.5 translates into 1.3%g. Any ground motion thresholds higher than that, while permitted, are actively advised against due to timeliness requirements. The ShakeAlert approach of tolerating some degree of over-alerting/bias in its ground motion predictions near the epicenter is not alone. For instance, the Japanese EEW system alerts entire subprefectures regardless of the range of ground motions predicted across those individual geographic polygons. Moreover, the Japanese system has implemented an algorithm (PLUM) that is specifically designed to systematically overpredict intensity levels (<https://doi.org/10.1186/s40623-025-02172-2>). While EEW systems are all a little bit different, the approach envisioned in this paper for first alerts based on DAS data, is very much in line with the current practice of the ShakeAlert and Japanese EEW systems. It is similar to their prioritizing speed over ground motion accuracy near the

epicenter once a significant earthquake is detected. Given the difficulties locating earthquakes using a relatively straight (1D) fiber path offshore, we consider this a viable approach.

- 2. Given that the five most significant features are highly correlated (Fig. S2, with three of them being 100% correlated). Why not removing these attributes to improve the predictions?**

Indeed, removing the three correlated (and lower ranking) features has improved the precision metric of predictions, and, we believe, is a sensible step to add to the method; thank you for your suggestion. Below are two figures: the top one is added to the supplement displaying the 3 remaining, less correlated figures, and the bottom is the resultant outcome. The precision score – which, to us, is the most important metric – has improved by 2%. We've adjusted the text accordingly (Results; paragraph 2).

Figure 5. ...The top 6 features (left of dashed fuchsia line) are determined from the second round of XGBoost, then the 3 features which are less correlated with each other (those labeled with red text) are used in the final predictive model....

Minor comments and questions:

1. In the abstract: "shows high precision compared to real-time EEW systems." Specify which EEWS you have compared with; in this case, it is ShakeAlert. The reading flow even starts with ShakeAlert in the introduction.

Now: "the EEW system ShakeAlert®,"

2. Line 93: Perhaps add ref. Meier et. al 2017 (<https://doi.org/10.1126/science.aan5643>), because the Source Time Function for large earthquakes is typically < 6 s.

This M7.0 event was found to have an approximately 10-15 s source time function:

Pollitz, F. F., Guns, K. A., & Yoon, C. E. (2025). Rupture process of the Mw7.0 December 5, 2024 Offshore Cape Mendocino earthquake. *Geophysical Research Letters*, 52, e2025GL115613. <https://doi.org/10.1029/2025GL115613>

3. Line 96-99: Which hyperparameters were used for the Decision Tree and XGB models? For example, number of trees, maximum depth, percentage of the initial feature vector, etc. You might also consider adding a reference that uses a single station, XGB, and 3 seconds of the P-wave (<https://doi.org/10.1029/2023JB026575>).

N-estimators is the number of trees (now clearly specified in supplementary material). Maximum depth is in the supplementary material tables S4 , and we've now added the percentage of the initial feature vector to Supplementary material Table S4.

Cited Lara et al., 2023 (paper mentioned above) in Discussion, paragraph 3:

"Additionally, (Lara et al., 2023) used XGBoost on statistics derived from 3 s of earthquake waveform data from single seismometers to predict magnitudes from global M3-M9 earthquakes, with the accuracy of results decreasing as magnitudes increased beyond M6."

4. In the feature importance analysis, I wonder if the SHAP scores are highly correlated with the feature importance values provided by the XGB model.

We've added this figure to the supplement, and the text below the figure caption to the manuscript:

Figure S. Comparison of SHAP feature importance score (blue) and XGBoost gain (orange), both normalized from zero to one. XGBoost gain describes how well each feature improved a given split, averaged over all splits in the model. SHAP scores, on the other hand, describe how a given feature impacts the prediction overall.

3rd paragraph in section 4.4:

"While XGBoost's internal gain metric indicates how much a feature improves the model's splits during training, it primarily reflects the feature's role in model construction rather than its direct influence on predictions. In contrast, SHAP values attribute changes in predicted scores to specific features, providing a more consistent and interpretable measure of their overall predictive contribution. For this reason, we use SHAP scores to evaluate feature importance."

5. Line 133: It is unclear what exactly constitutes these 500 models. Are these 500 models trained with different parameters? Please clarify whether this refers to 500 independent training runs (e.g., via cross-validation, bootstrapping, or parameter variation).

The models with the 56 features had the grid search with K-folds cross validation, the 500-epoch-models with 6 (now 3) features use the optimal parameters found from the 56-feature models. I believe we've clarified this now:

Section 2.1, 2nd paragraph:

“After determining the most important features and optimal hyperparameters from the 56-feature models, we then systematically test between 1 and 1000 epochs to train XGBoost, finding that 500 epochs lead to the optimal results. Here, all training and testing are performed on borehole strainmeter data.”

Section 4.4, 3rd paragraph:

“For the initial model training round of XGBoost with 56 features, we grid-search with K-folds cross-validation to find the optimal hyperparameters for our models, details about which can be found in the Supplementary Material Table S4.”

6. Line 160: True Positive (TP) is defined as $M > 5.4$ earthquakes estimated as $M > 5.4$, FP as $M < 5.4$ earthquakes estimated as $M > 5.4$, and FN as $M > 5.4$ earthquakes estimated as $M < 5.4$. The strong point is the larger number of $M > 5.4$ samples used in the proposed algorithm compared to only 3 in ShakeAlert. The precision and recall for these data indicate that the proposed algorithm has a lower chance of producing an FP. However, in an EEW, the critical issue is the algorithm's failure in the case of an FN.

It depends on end-user risk tolerance and the intended goal of the alerting system: if there are too many false positives then people who receive alerts could develop alert fatigue, and then either ignore future alerts or uninstall the MyShake app that sends alerts, or opt out of all Wireless Emergency Alerts. Roughly 10% of the U.S. population has already opted out of WEAs. ShakeAlert currently has this problem, as described in the text, so we are trying to mitigate this issue. Also, the automated responses for a major earthquake (sending wireless emergency alerts, elevator and trains stopping, etc.) can be highly disruptive for false positives. Finally, it is unlikely that the existing ShakeAlert network will have FNs in the event of a large earthquake, so the addition of DAS is meant to bolster the existing

network, and in that case, missing a large earthquake is not as disruptive as alerting for a small earthquake.

Response to Reviewers

Rapid earthquake magnitude classification via P-wave strains from borehole strainmeters and Distributed Acoustic Sensing;
Sawi et al.

We thank the reviewer for their time and their thorough review of our manuscript. We appreciate the helpful feedback and thoughtful suggestions, which have improved this work considerably. We do, however, disagree with the reviewer's opinions regarding the impact and contribution that this work would bring to Nature Communications. We believe that this work represents a considerable advance in the understanding of the importance of low-frequency strain information for characterizing large-magnitude earthquakes early in their onset, with implications for future or operational earthquake early warning algorithms. This entire project was funded by the ShakeAlert R&D effort because USGS has identified DAS based P-wave magnitude estimates as the key scientific step in operationalizing DAS in ShakeAlert. McGuire serves as the ShakeAlert chief scientist, and our study is designed to directly impact the operational EEW system in the near future. The model's transferability to DAS data further increases the broad impact of this work, as the field of DAS seismology — and especially its application for earthquake early warning — becomes more widespread.

We have addressed the reviewer's comments below. Their unedited comments are in black, and our responses are in blue.

Dear Authors, dear Editors,

I have read this manuscript with great interest. I think this is an excellent and scientifically very robust manuscript that unquestionably deserves publication. Its technical quality, clarity, and rigor are high, and the results represent a meaningful contribution to the field. However, as currently written, the manuscript does not articulate the kind of broad, field-shaping impact expected for Nature Communications. The work may be more appropriately aligned with a high-level disciplinary journal (such as Nature Geoscience, Communications Earth & Environment, or GRL), where its strengths would be fully appreciated.

The methodological approach relies on established techniques applied to a familiar classification problem, and the central findings, while interesting, do not emerge as a clear conceptual advance that would reshape current understanding or practice. The physical interpretation remains qualitative, and the potential implications for early source characterization are not explored in a way that convincingly highlights a transformative contribution.

Our model results confirm that the frequency content of early arrivals has predictive power on the source magnitude. We argue that this result *should* inform the current practice of operational EEW systems, which rely primarily on waveform amplitudes — not frequency content — for magnitude estimations. This finding is unique in that we are leveraging the the ultra-broadband sensitivity of the strainmeters to demonstrate that low frequency signals inform source magnitude within seconds of the P-wave — a concept that, until this study, has not been demonstrated at such a scale.

Additionally, no study has tested DAS strain against independent strainmeter systems to the extent that his study has. Being able to successfully apply an ML model trained on strainmeter data to DAS data is a remarkable finding. It moves the needle closer to widespread adoption of DAS as an accepted seismological instrument. This is particularly important as DAS data on standard telecom cables, even in offshore settings, is unlikely to record the high-amplitude strains during large earthquake S-waves with fidelity as the sensing medium (e.g. the fiber) is not rigidly attached to solid rock. Indeed, we see this with the M7.0 earthquake DAS data used in this paper which only experienced MMI 5 shaking. JAMSTEC demonstrated that seafloor cables do not provide reliable accelerograms in M8 megathrust earthquakes 20+ years ago. This inherent limit of fiber-optics implies the need to maximize the information content of the relatively well recorded, low-amplitude, P-wave portion of the waveform. Hence, our new method is directly addressed at a widespread, global problem that is inherent in all DAS data that is not derived from fiber cemented into solid rock.

The discussion of DAS-based applications would benefit from a more realistic appraisal of current operational limitations. DAS remains extremely challenging to process in real time at the scale required for EEW, and to date, to my knowledge, it is not used operationally for monitoring precisely because of the computational burden and data volume. Without demonstrating real-time feasibility or proposing a strategy to overcome these constraints, it is difficult to evaluate the practical significance of the DAS extension.

Gou et al., (2025, *Scientific Reports*) and Biondi et al. (2025; arXiv preprint) present real-time DAS-integrated EEW-frameworks that perform with operational capabilities, so it is indeed feasible to include DAS data in EEW systems. We have added to the Discussion section (bottom of page 9/top of page 10) to more clearly propose a strategy, and mention work already performed, towards real-time deployment of this method. Additionally, DAS data is already used operationally by Caltech in the Southern California Seismic Network already for arrival time information. The USGS led Advanced National Seismic System has a working group focused on implementing DAS in real-time monitoring operations on a nationwide scale. It is only a matter of time before it becomes part of ShakeAlert.

A second concern relates to the question of model transferability. The manuscript focuses on a single region and dataset, yet does not explore whether the proposed features or learned representations would generalize to other tectonic contexts, sensor networks, or noise environments.

The strainmeter dataset comes from all over California, with different tectonic regimes including a triple junction with both relatively deep intraslab and transform fault events at the Mendocino Triple Junction, and crustal strike-slip events in Napa and Ridgecrest. The only tectonic region we have left out is basin and range normal faults.

The same, or similar, strainmeter instruments are used in geophysical networks across the globe, strongly suggesting that our simple method of calculating statistical features (e.g., order of magnitude of maximum amplitude or bandpassed wavelet transform) would be generalizable to their data as well.

These three regions (MTJ, Napa, and Ridgecrest) presumably experience different noise environments naturally by being in widely different locales and geologic regions. The MTJ sensors, and our DAS data, being located close to the coast in the Northwest Pacific experience high levels of ocean generated noise, particularly in the winter. Our DAS dataset experiences huge amounts of traffic noise during daytime/commuting hours, but is very quiet at night, providing a range of conditions for the data in Figure 8.

The positioning of the work would also benefit from a deeper engagement with the most recent literature. There is a notable gap in the manuscript's coverage of the broader Earthquake Early Warning (EEW) literature, particularly work that explicitly targets rapid magnitude estimation and waveform classification using machine-learning approaches. The authors do not sufficiently contrast their findings with key recent studies (e.g., the line of work following Lomax et al., 2019, and related developments) nor with methodological advances coming from adjacent fields, such as rapid frequency-domain classifiers or approaches inspired by gravity-waves signal processing. It would strengthen the paper significantly if the authors (i) positioned their contribution within these bodies of work, (ii) clarified what is genuinely new in terms of methodology or physical insight compared with these precedents, and (iii) provided direct comparisons when possible or justified their absence.

We have now added text in the first 2 paragraphs and last paragraph of the Discussion section:

1. Comparisons of our study to recent works on machine learning for rapid magnitude estimation for EEW (Joshi et al., 2024; Hou et al., 2024; Zhu et al., 2025). Joshi et al., used a similar method to ours (XGBoost + SHAP importance metric), but filtered their waveforms above 1 Hz, thus preventing them from discovering low frequency characteristics of the large-M earthquakes. Hou and Zhu used deep learning networks, obscuring which features are most important, preventing the reader from knowing whether amplitude- or frequency- based features are most predictive. ^[SEP]
2. Limitations of speed from early works on magnitude characterization (Lomax et al., 2019; Mousavi et al., 2020) ^[SEP]
3. Describing the rapid frequency-domain classifier, Variation Mode Decomposition (Liu et al., 2024; Liu et al., 2025), and its comparison to continuous wavelet transforms ^[SEP]
4. And describing how our work could compliment prompt elastogravity signals of very large earthquakes in an EEW context (Liccardi et al., 2022). Elastogravity signals are unlikely to be part of operational EEW in the near future due to the extreme expense of operating the sensing instruments and the long time intervals between truly great earthquakes in a given country.

I think that this study, while promising, does not yet demonstrate the level of maturity expected for a Nature Communications contribution.

Showing that the approach can scale beyond a single region,

Our strainmeter dataset comes from different tectonic regimes (crustal strike-slip, transform faulting, subducting slab), in different geologic and noise environments, and spread across large geographic distances, so I believe we demonstrate that our method is generalizable beyond a single region.

operate in real-time conditions,

To demonstrate operation in real time, we would optimize the the algorithms to minimize computation time, then run them in parallel on GPUs on-site at the cable — this is beyond the scope of this work, as we now mention in the text. However, GPU based algorithms are running on site in California DAS systems already [Gou et al., 2025 (Scientific Reports)] and that approach is likely to become part of ShakeAlert in the near

future. We have suggested plans to overcome barriers to implementation in real-time and cited a work that presents DAS integration into an operational EEW system. We believe this is sufficient for this study.

and consistently perform on a broader set of events

The model performs very well on our test data set, which do come from a broad set of events. They range from M3.5 to 7.1, spanning different tectonic and geologic settings, from regions 100s of km apart. The same, or similar, strainmeters are deployed elsewhere, and there is no reason to believe that the statistical features (e.g., order of magnitude of strain amplitude or power of continuous wavelet transform) of waveforms from those instruments would vary widely.

...would considerably strengthen the case for publication in a journal with the wide reach and expectations of Nature Communications. Once such demonstrations are in place, the work may well achieve the level of generality and impact required for that venue.

Dr. Theresa Sawi

February 23, 2026

Response to Reviewers, NCOMMS-25-21590B

We thank the reviewer for taking the time to read through our revisions and for the thoughtful comments they have provided. The reviewer's comments are reprinted here verbatim in black, and our responses are in blue:

Thank you for your interest in my opinion regarding the critical points raised. I feel that the authors have adequately addressed the comment concerning model transferability. However, I would like to offer one additional suggestion that could further strengthen the discussion of real-time feasibility.

The authors have now added references to DAS-based EEW systems in the Discussion section, which reinforces the case for future implementation of their methodology. It may be helpful to explicitly mention that the USGS is actively working toward implementing DAS in real time on a nationwide scale, so that readers clearly understand that DAS-based EEW is likely to become an operational reality.

Thank you for this comment. Given that seismic hazards tend to be greatest around the West Coast and Alaska regions, this is where the initial phase of operational DAS for earthquake early warning is centralized. That is, I'm not sure that emphasizing a "nation-wide scale" is appropriate in this context. DAS-based EEW is already being deployed onsite in California (e.g., Gou et al., 2025; Scientific Reports).

To further support the feasibility of the proposed methodology within an EEW framework, the authors could consider including an estimate of the algorithm's time complexity, excluding data transmission and telemetry delays. For example, this would consist of the required 4 seconds of P-wave data plus the computational time needed for feature extraction and classification. Even more compelling would be to provide an estimate of the expected user lead time, calculated as the difference between the theoretical S-wave arrival time (or the time corresponding to observed 5%g shaking) at a given location and the time at which the algorithm classifies the event as large (see the Discussion section on the "late-alert zone" in <https://doi.org/10.1785/0120240119>).

As we mention in our previous "response to reviewers," providing an accurate measure of time complexity would not be possible without utilizing GPUs at the site of the

interrogator and optimizing the algorithm to minimize computational time, which is beyond the scope of this study.

In the first paragraph of the Discussion, the authors may also consider citing reference (49), Lara et al. (2023), including its real-time application (<https://doi.org/10.1785/0120240119>), as this methodology likewise estimates large earthquakes using XGBoost with waveforms filtered above 1 Hz.

We have cited Lara et al. (2023) in the first paragraph of the Discussion as suggested, as the method is relevant. Since the method is the same in Lara et al., (2025), we have not cited that paper, interesting though it is.

I would recommend acceptance of the manuscript in its current form. Nevertheless, I believe that incorporating these clarifications would further strengthen the paper, particularly with respect to the real-time feasibility considerations raised during the review process.

Thank you for your review and your recommendation.